# Dynamics of Semiconductor Lasers under External Optical Feedback from Both Sides of the Laser Cavity

**Mónica Far Brusatori * and Nicolas Volet**

Department of Electrical and Computer Engineering, Aarhus University, 8200 Aarhus, Denmark; volet@ece.au.dk
* Correspondence: mfar@ece.au.dk

**Abstract:** To increase the spectral efficiency of coherent communication systems, lasers with ever-narrower linewidths are required as they enable higher-order modulation formats with lower bit-error rates. In particular, semiconductor lasers are a key component due to their compactness, low power consumption, and potential for mass production. In field-testing scenarios their output is coupled to a fiber, making them susceptible to external optical feedback (EOF). This has a detrimental effect on its stability, thus it is traditionally countered by employing, for example, optical isolators and angled output waveguides. In this work, EOF is explored in a novel way with the aim to reduce and stabilize the laser linewidth. EOF has been traditionally studied in the case where it is applied to only one side of the laser cavity. In contrast, this work gives a generalization to the case of feedback on both sides. It is implemented using photonic components available via generic foundry platforms, thus creating a path towards devices with high technology-readiness level. Numerical results shows an improvement in performance of the double-feedback case with respect to the single-feedback case. In particularly, by appropriately selecting the phase of the feedback from both sides, a broad stability regime is discovered. This work paves the way towards low-cost, integrated and stable narrow-linewidth integrated lasers.

**Keywords:** laser dynamics; optical feedback; narrow-linewidth lasers; semiconductor lasers; laser stability

## 1. Introduction

The effect of external optical feedback (EOF) on diode laser dynamics has been extensively studied for the past half century [1,2]. EOF has been proven to affect laser performance, showing regimes that can aid in linewidth reduction [3–6], as well as others responsible for highly unstable behavior, from mode hopping to the case of coherence collapse [7–12]. Methods to improve laser stability thus need to take EOF into account, as even weak feedback can be detrimental. A traditional approach to mitigate its effects is to include an off-chip isolator at the laser output. Yet, this component negatively impacts the dimensions of packaged devices as well as fabrication times and costs. As such, research is ongoing to develop an integrated solution that can minimize the negative effects of EOF. Efforts include adjusting the feedback phase to tune into line-narrowing regimes [13,14], using unidirectional phase modulators [15,16], reducing the linewidth enhancement factor, e.g., using quantum dots [17–19], employing electromagnetic effects [20,21], harnessing the mode propagation properties of ring lasers [22,23], or the extended cavity approach [24–27].

Double external feedback on the same side of the laser cavity has been previously explored for chaos stabilization in Reference [28]. Its results numerically show that, by introducing an additional feedback term, a chaotic regime can be stabilized in terms of output power, where a robust stable region for a wide parameter range can be seen. Furthermore, linewidth is numerically shown to be reduced with respect to the single feedback case. However, it lacks an analytical expression for the linewidth, and it does not explore the effect of the added term on other optical feedback regimes, which limits the scope of the analysis [7]. Dual external cavities have also been explored to maximize the sensitivity

of self-mixing interferometers for sensing applications [29–31]. These studies focus on improving the interferometric signal in terms of output power characteristics, yet they do not study the effect EOF has on linewidth, which is crucial for other applications such as coherent communications [32] or frequency metrology [33]. Furthermore, in all works exploring dual external cavities, both external reflections return to the same side of the laser cavity and thus have the same propagation direction. Moreover, phase shifts in the external phase cavity are not accounted for, thus ignoring, e.g., possible phase shifts at the mirrors.

The established line of thought relies on the key assumption that feedback is introduced from only one side of the laser cavity. Current integration technologies make this assumption obsolete, as they allow for arbitrarily complex design geometries with a variety of functionalities, such as tunability and modulation, while maintaining narrow-linewidth performance [34–38]. Consequently, this work aims to extend the theoretical foundations of EOF to the case of feedback coupling into the laser cavity from both sides. This system is studied to obtain and analyze its dynamic rate equations. Furthermore, the frequency noise power spectral density and subsequently the intrinsic linewidth's dependence on feedback is computed. The Lang–Kobayashi approach is used [39], where an additional term to account for the extra feedback term is included. In a similar fashion, to obtain analytical solutions both small-signal and weak feedback conditions are assumed. The obtained equations are then numerically solved. Results show that feedback-insensitivity can be achieved by tuning the feedback parameters. In contrast with previous works, this can be accomplished in a monolithic platform by including a single active component in a straightforward geometric configuration. As such, design complexity is reduced which enables devices with a compact form factor. Furthermore, existing methods for laser fabrication are suitable for realizing the proposed device. In particular, the proposed laser system can be made with mature photonic integrated components available in generic foundry platforms [40,41], thus creating a path towards devices with high technology-readiness level while maintaining low cost and size.

## 2. Rate Equations Model

This section includes a derivation of the dynamic equations of a laser cavity with EOF coupled into the laser cavity from both sides. Starting from the Lang–Kobayashi model [39], the lasing frequency and threshold gain shifts due to feedback are obtained, as well as an analytical expression for the frequency noise power spectral density from which the change of intrinsic linewidth can be computed.

This work proposes a revised laser system, as shown in Figure 1. The laser cavity of length $L$ is delimited by two mirrors with complex reflection coefficients $\rho_1$ and $\rho_2$, respectively. Assuming two interfaces at each side of the main cavity, two additional back-reflections ($\rho_{1,\text{ext}}$ and $\rho_{2,\text{ext}}$) are considered and accounted for by computing effective reflection coefficients. The case of weak feedback is discussed here, for which:

$$\left|\rho_{j,\text{ext}}\right| \ll 1. \tag{1}$$

The following parameters are introduced:

$$\kappa_j \equiv \frac{1 - \rho_j^2}{\rho_j} \frac{\left|\rho_{j,\text{ext}}\right|}{t_{\text{cav}}} \tag{2a}$$

$$\phi_j \equiv \omega_{\text{FB}} t_j + \phi_{\text{m}_j} \tag{2b}$$

$$\delta\omega \equiv \omega_{\text{FB}} - \omega_{\text{ref}} \tag{2c}$$

$$t_{\text{cav}} = 2L/v_{\text{g}}, \tag{2d}$$

with $j = 1, 2$, where $\kappa_j$ is the coupling coefficient; $t_{\text{cav}}$ is the cavity round-trip time, with the group velocity $v_{\text{g}}$; $\phi_j$ is the phase delay due to the external cavities determined by the

external round-trip time $t_j$, the lasing frequency in the presence of feedback $\omega_{\mathrm{FB}}$, and a phase shift at the external mirrors $\phi_{\mathrm{m}_j}$; and $\omega_{\mathrm{ref}}$ is the free running laser frequency.

The first step is extracting the lasing conditions of the proposed laser system.

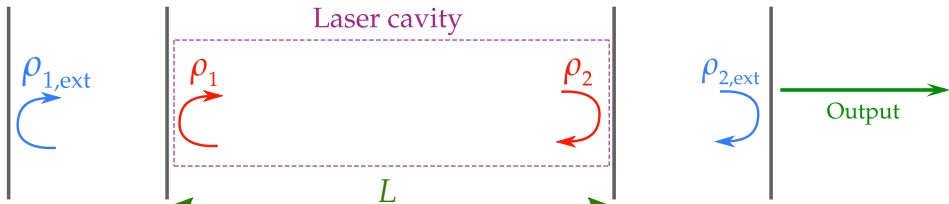

**Figure 1.** Schematic of a laser cavity affected by external optical feedback from both sides.

*2.1. Lasing Conditions*

By analyzing the slowly-varying amplitude of the complex electric field, effective reflection coefficients $\left(\rho_j^{\mathrm{eff}}\right)$ can be obtained:

$$\frac{\rho_j^{\mathrm{eff}}}{\rho_j} \approx 1 + \kappa_j t_{\mathrm{cav}} e^{\pm i\phi_j}, \tag{3}$$

where the plus and minus signs corresponds to $j = 1$ (left mirror) and $j = 2$ (right mirror), respectively. The additional reflection influences the lasing condition, as shown in Appendix A, and thus the threshold gain and lasing frequency. To calculate them, the following definitions are convenient:

$$G_{\mathrm{FB}} \equiv \Gamma g_{\mathrm{FB}} v_{\mathrm{g}} \tag{4a}$$

$$G_{\mathrm{ref}} \equiv \Gamma g_{\mathrm{ref}} v_{\mathrm{g}} \tag{4b}$$

$$\Delta\phi_{\mathrm{m}} \equiv \phi_{\mathrm{m}_1} - \phi_{\mathrm{m}_2}, \tag{4c}$$

$$\gamma_{\mathrm{H}} \equiv \sqrt{1 + \alpha_{\mathrm{H}}^2} \tag{4d}$$

$$\theta_{\mathrm{H}} \equiv \arctan(\alpha_{\mathrm{H}}) \tag{4e}$$

where $\Gamma$ is the confinement factor; $g_{\mathrm{FB}}$ and $g_{\mathrm{ref}}$ are the threshold gain coefficient with and without feedback, respectively; and $\alpha_{\mathrm{H}}$ is the linewidth enhancement factor [42]. From Appendix B, it is possible to obtain:

$$G_{\mathrm{FB}} - G_{\mathrm{ref}} \equiv \delta G \overset{(\mathrm{A}16)}{\approx} -2\kappa_2 \cos(\phi_2) - 2\kappa_1 \cos(\phi_1) \tag{5a}$$

$$\delta\omega \overset{(\mathrm{A}23)}{\approx} -\gamma_{\mathrm{H}}[\kappa_2 \sin(\phi_2 + \theta_{\mathrm{H}}) + \kappa_1 \sin(\theta_{\mathrm{H}} - \phi_1)]. \tag{5b}$$

The relation between the right-hand terms determines the shift in threshold gain and lasing frequency with feedback. When comparing with previous literature regarding dual external cavity lasers, a sign difference is observed with respect to the lasing frequency shift equations found in References [28,29], arising from the different propagation direction of the additional feedback term. A clearer contrast is present with the lasing frequency shift shown in ref. [30], where a single term accounts for both external reflections. Furthermore, a phase shift at the external mirrors ($\phi_{\mathrm{m}_j}$) is not considered in the mentioned sources, thus the feedback terms included in both the excess gain and lasing frequency variations are distinct with respect to those in Equation (5).

Given the transcendental form of Equation (5b), a numerical analysis under different feedback conditions is studied in Section 3. Nevertheless, an analytical solution for lasing frequency stability can be found for the condition:

$$\delta\omega = 0. \tag{6}$$

Under this condition, Equation (5b) becomes:

$$\kappa_2 \sin(\phi_2 + \theta_{\mathrm{H}}) \overset{(6)}{=} -\kappa_1 \sin(\theta_{\mathrm{H}} - \phi_1), \tag{7}$$

which is determined by the feedback parameters $\kappa_j$ and $\phi_j$, the latter being dependent on the time delay $t_j$ as well as the lasing frequency. Finding a stable solution that does not depend on the lasing frequency is of particular interest, as it can be advantageous for tunable lasers and their numerous applications. With the following assumption:

$$\kappa_2 = \kappa_1 \equiv \kappa, \tag{8}$$

Equation (7) can be rewritten as:

$$\sin(\phi_2 + \theta_{\mathrm{H}}) = -\sin(\theta_{\mathrm{H}} - \phi_1) = \sin(\phi_1 - \theta_{\mathrm{H}})$$
$$\Rightarrow \phi_2 + \theta_{\mathrm{H}} = \phi_1 - \theta_{\mathrm{H}} + 2m\pi. \tag{9}$$

Without loss of generality, the parameter $m$ is set to $m = 0$, thus:

$$2\theta_{\mathrm{H}} = \phi_1 - \phi_2 \overset{(2b)(4c)}{=} \omega_{\mathrm{FB}}(t_1 - t_2) + \Delta\phi_{\mathrm{m}}$$
$$\Rightarrow 2\theta_{\mathrm{H}} - \Delta\phi_{\mathrm{m}} = \omega_{\mathrm{FB}}(t_1 - t_2). \tag{10}$$

Choosing an equal time delay in both external cavities:

$$t_2 = t_1 \equiv t_{\mathrm{ext}}, \tag{11}$$

sets the right hand term of Equation (10) to zero, so that:

$$2\theta_{\mathrm{H}} = \Delta\phi_{\mathrm{m}}. \tag{12}$$

This result suggests that by tuning the phase in the external cavities, so that condition (12) is met, it is possible to obtain a feedback-insensitive lasing frequency. An active method to tune the phase is however required as $\alpha_{\mathrm{H}}$ is dependent on laser parameters, such as carrier density and wavelength [43]. This can be managed by, for example, phase shifters, which are mature and widely used components that can be included on-chip in a laser. The shown stable solution thus requires meeting the conditions (8), (11) and (12), which constrain the feedback parameters of one side of the cavity with respect to those of the other side, but do not restrict their absolute value. Nevertheless, if conditions (8) and (11) are not met, solutions for stable performance become frequency dependent. Such a case would thus only be satisfied for certain lasing frequency values for a given set of feedback parameters, which can potentially yield unstable solutions for other frequencies.

Regarding the variations in threshold gain, under conditions (8) and (11) Equation (5a) becomes:

$$\delta G = -2\kappa[\cos(\omega_{\mathrm{ref}} t_{\mathrm{ext}} + \phi_{\mathrm{m}_1}) + \cos(\omega_{\mathrm{ref}} t_{\mathrm{ext}} + \phi_{\mathrm{m}_2}))], \tag{13}$$

which vanishes if:

$$\cos(\omega_{\text{ref}} t_{\text{ext}} + \phi_{m_1}) = -\cos(\omega_{\text{ref}} t_{\text{ext}} + \phi_{m_2})$$

$$\cos(\omega_{\text{ref}} t_{\text{ext}} + \phi_{m_1}) = \cos(\omega_{\text{ref}} t_{\text{ext}} + \phi_{m_2} - \pi) \tag{14}$$

$$\Rightarrow \omega_{\text{ref}} t_{\text{ext}} + \phi_{m_1} = \omega_{\text{ref}} t_{\text{ext}} + \phi_{m_2} - \pi + 2m\pi \quad m \in \mathbb{Z}$$

$$\overset{m=0}{\Rightarrow} \Delta\phi_m = \pi.$$

This condition, while different than condition (12), also yields stability regardless of lasing frequency in this case for the threshold gain. Both cases are studied numerically in Section 3.

### 2.2. Rate Equations

In order to obtain the frequency noise (FN) power spectral density (PSD), and from it the laser intrinsic linewidth, the laser rate equations for the intensity and phase, as well as one for the carrier number, need to be studied. The former two can be extracted from the dynamic equations for the field inside the laser cavity, following the Lang-Kobayashi [39] approach. Its full derivation is shown in Appendix C. Furthermore, Langevin noise terms are included to account for shot noise fluctuations. The following definitions are useful to simplify notation:

$$\mathcal{A}(t) = \sqrt{S(t)} e^{-i\phi(t)} \tag{15a}$$

$$\mathcal{S}_j^\pm \equiv \kappa_j \sqrt{S(t \pm t_j)} \tag{15b}$$

$$\phi_{t_j}^\pm \equiv \phi(t \pm t_j) \tag{15c}$$

$$\Delta\Phi_1^+ \overset{(15c)}{=} \phi(t) - \phi_{t_1}^+ - \phi_1 \tag{15d}$$

$$\Delta\Phi_2^- \overset{(15c)}{=} \phi(t) - \phi_{t_2}^- + \phi_2 \tag{15e}$$

$$\Delta G \equiv G_{\text{FB}} - \tau_{\text{ph}}^{-1}, \tag{15f}$$

with $j = 1, 2$ relating to the EOF components from the left and right, respectively, and $\tau_{\text{ph}}$ is the photon decay time, which accounts for cavity and mirror losses. The field amplitude $\mathcal{A}$ is assumed to be slowly varying, where $S$ is the photon number inside the laser cavity and $\phi$ is the phase of the field. With these definitions, the rate equations of the system can be written as:

$$\dot{S} \overset{(A27)}{=} S\Delta G + 2\mathcal{S}_2^- \sqrt{S} \cos(\Delta\Phi_2^-) + 2\mathcal{S}_1^+ \sqrt{S} \cos(\Delta\Phi_1^+) + R_{\text{sp}} + F_S \tag{16a}$$

$$\dot{\phi} \overset{(A28)}{=} \frac{\alpha_H \Delta G}{2} - \delta\omega - \frac{\mathcal{S}_2^-}{\sqrt{S}} \sin(\Delta\Phi_2^-) - \frac{\mathcal{S}_1^+}{\sqrt{S}} \sin(\Delta\Phi_1^+) + F_\phi \tag{16b}$$

$$\dot{N} \overset{[3]}{=} I - GS(t) - N\tau_{\text{sp}}^{-1} + F_N, \tag{16c}$$

where $I$ is the effective rate of injected current (in electrons), $\tau_{\text{sp}}$ is the carrier lifetime, and $R_{\text{sp}}$ is the spontaneous recombination rate. This system is compatible with those presented in refs. [28,31], by accounting for the sign change in the delayed feedback term corresponding to the left external mirror, due to it having the opposite propagation direction.

The steady state solution of Equation (16) can be seen in Equation (A33). The Langevin noise sources $F_S(t)$, $F_\phi(t)$ and $F_N(t)$ satisfy [44]:

$$\langle F_i(t) \rangle = 0 \tag{17a}$$

$$\langle F_i(t_1)F_j(t_2) \rangle = 2D_{ij}\delta(t_1 - t_2) \quad \text{with } i, j = S, \phi \text{ or } N, \tag{17b}$$

where:

$$D_{SS} = R_{\text{sp}}S \quad ; \quad D_{\phi\phi} = \frac{R_{\text{sp}}}{4S} \quad ; \quad D_{NN} = R_{\text{sp}}S + N\tau_{\text{sp}}^{-1} \quad ; \quad D_{SN} = -R_{\text{sp}}S, \tag{18}$$

are standard diffusion coefficients. A usual approach for solving the system from Equation (16) involves a small-signal analysis. Small deviations from a steady-state value are assumed:

$$S \simeq S_0 + S_\Delta = S_0 + \int_{-\infty}^{\infty} e^{i\Omega't}S_{0\text{p}}(\Omega')\mathrm{d}\Omega' \quad \text{with} \quad S_0 \gg S_\Delta \tag{19a}$$

$$\phi \simeq \phi_\Delta = \int_{-\infty}^{\infty} e^{i\Omega't}\phi_{0\text{p}}(\Omega')\mathrm{d}\Omega' \tag{19b}$$

$$N \simeq N_0 + N_\Delta = N_0 + \int_{-\infty}^{\infty} e^{i\Omega't}N_{0\text{p}}(\Omega')\mathrm{d}\Omega' \quad \text{with} \quad N_0 \gg N_\Delta, \tag{19c}$$

where the steady state value of the phase is assumed to be zero. The full linearization of the rate equations is shown in Appendix D, which uses the following definitions:

$$\kappa_j^{\text{c}} \equiv \kappa_j t_j \cos(\phi_j) \tag{20a}$$

$$\kappa_j^{\text{s}} \equiv \kappa_j t_j \sin(\phi_j) \tag{20b}$$

$$\mathsf{K}_s \equiv \kappa_2^{\text{s}} + \kappa_1^{\text{s}} \tag{20c}$$

$$\mathsf{K}_c \equiv 1 + \kappa_2^{\text{c}} - \kappa_1^{\text{c}} \tag{20d}$$

$$\zeta_s \equiv R_{\text{sp}}/S_0 \tag{20e}$$

$$a_g = \Gamma v_g a \tag{20f}$$

$$G_i \approx a_g(N_i - N_{\text{tr}}) \tag{20g}$$

$$\tau_{\text{e}}^{-1} \equiv a_g S_0 + \tau_{\text{sp}}^{-1}, \tag{20h}$$

where a linear approximation for the gain has been introduced, with $a$ the differential gain coefficient and $N_{\text{tr}}$ the number of electrons at transparency. Applying the Fourier transform to Equation (A36a–c), the following system of equations is obtained in the frequency domain:

$$i\Omega\mathsf{K}_c S_{0\text{p}} \overset{(A36a)}{=} a_g S_0 N_{0\text{p}} - \zeta_s S_{0\text{p}} - 2i\Omega S_0 \mathsf{K}_s \phi_{0\text{p}} + \widehat{F}_S \tag{21a}$$

$$2i\Omega\mathsf{K}_c\phi_{0\text{p}} \overset{(A36b)}{=} \alpha_{\text{H}} a_g N_{0\text{p}} + i\Omega\frac{\mathsf{K}_s}{S_0}S_{0\text{p}} + 2\widehat{F}_\phi \tag{21b}$$

$$i\Omega N_{0\text{p}} \overset{(A36c)}{=} -\tau_{\text{e}}^{-1}N_{0\text{p}} - G_0 S_{0\text{p}} + \widehat{F}_N, \tag{21c}$$

where the unknowns $S_{0\text{p}}$, $\phi_{0\text{p}}$, $N_{0\text{p}}$, and $\widehat{F}_S$, $\widehat{F}_\phi$ and $\widehat{F}_N$ depend on the Fourier angular frequency $\Omega$. These equations are the first step to obtain the FN PSD.

### 2.3. Power Spectral Density and Laser Intrinsic Linewidth

The next step is to calculate the FN PSD, and from it the laser intrinsic linewidth. By solving the system in Equation (21) it is possible to find an expression for the PSD [45]:

$$S_f^{(1)}(\Omega) = \frac{\Omega^2}{2\pi^2}\langle|\phi_{0\mathrm{p}}(\Omega)|^2\rangle, \tag{22}$$

which, using the following definitions:

$$\mathsf{F}_0 \equiv \zeta_s^2 + \mathsf{K}_c^2\tau_\mathrm{e}^{-2} - 2\mathsf{K}_c a_g G_0 S_0 \tag{23a}$$

$$\mathsf{F}_1 \equiv \mathsf{K}_s^2 + \mathsf{K}_c^2 \tag{23b}$$

$$\Lambda_4 \equiv 4\mathsf{F}_1 D_{\phi\phi} \tag{23c}$$

$$\mathsf{F}_2 \equiv \mathsf{K}_c\alpha_\mathrm{H} + \mathsf{K}_s \tag{23d}$$

$$\mathsf{F}_3 \equiv \mathsf{K}_c - \alpha_\mathrm{H}\mathsf{K}_s \tag{23e}$$

$$\Delta_4 \equiv \mathsf{F}_1^2, \tag{23f}$$

and:

$$\Lambda_2 \equiv a_g^2\mathsf{F}_2^2 D_{NN} + 4D_{\phi\phi}\left(\mathsf{K}_s^2\tau_\mathrm{e}^{-2} + \mathsf{F}_0\right) - 2\frac{\mathsf{K}_s}{S_0}D_{SS}a_g\left(\tau_\mathrm{e}^{-1}\mathsf{F}_2 - \zeta_s\alpha_\mathrm{H} - \alpha_\mathrm{H}G_0\right) \tag{23g}$$

$$\Lambda_0 \equiv \alpha_\mathrm{H}^2 a_g^2\left[D_{SS}\left(\zeta_s^2 + G_0^2 + 2\zeta_s G_0\right) + \zeta_s N\tau_\mathrm{sp}^{-1}\right] + \left(\tau_\mathrm{e}^{-1}\zeta_s + a_g G_0 S_0\right)^2 4D_{\phi\phi} \tag{23h}$$

$$\Delta_2 \equiv \mathsf{K}_c^2\zeta_s^2 + \tau_\mathrm{e}^{-2}\mathsf{F}_1^2 - 2\mathsf{F}_1 a_g G_0 S_0\mathsf{F}_3 \tag{23i}$$

$$\Delta_0 \equiv \left(a_g G_0 S_0\mathsf{F}_3 + \mathsf{K}_c\zeta_s\tau_\mathrm{e}^{-1}\right)^2, \tag{23j}$$

can be written as:

$$4\pi^2 S_f^{(1)} \stackrel{(23)(\mathrm{A}46)}{=\!=} \frac{\Lambda_4\Omega^4 + \Lambda_2\Omega^2 + \Lambda_0}{\Delta_4\Omega^4 + \Delta_2\Omega^2 + \Delta_0}. \tag{24}$$

This is an analytical solution for the FN PSD of the isolated laser system. From the following expression [46] :

$$S_f^{(1)}(\Omega \to 0) = 2\pi\Delta f, \tag{25}$$

which is valid for a Lorentzian lineshape, the intrinsic linewidth can be obtained. As shown in Appendix F:

$$F \equiv \frac{\Delta f}{\Delta f_0\left(1 + \alpha_\mathrm{H}^2\right)} \stackrel{(\mathrm{A}49)}{=} \left[1 + \gamma_\mathrm{H}\kappa_2 t_2\cos(\phi_2 + \theta_\mathrm{H}) - \gamma_\mathrm{H}\kappa_1 t_1\cos(\phi_1 - \theta_\mathrm{H})\right]^{-2}, \tag{26}$$

where $\Delta f_0$ is the Schawlow–Townes linewidth [47]. The expression found for the intrinsic linewidth has two feedback terms, one contribution from each side, with a sign that depends on $\phi_1$ and $\phi_2$. Recalling from Equation (2b) that these quantities are a function of $t_j$ and $\phi_{\mathrm{m}_j}$, a proper design of the laser can yield linewidth stability or a reduction of the intrinsic linewidth with respect to the case of one-sided feedback. This is further explored using a numerical analysis in Section 3. Additionally, it is possible to find an analytical expression

for Equation (26) under the conditions for frequency stability, namely conditions (8), (11) and (12). Using this assumptions in Equation (26):

$$F = \{1 + \gamma_{\mathrm{H}} \kappa t_{\mathrm{ext}}[\cos(\phi_2 + \theta_{\mathrm{H}}) - \cos(\phi_1 - \theta_{\mathrm{H}})]\}^{-2}. \tag{27}$$

Taking a closer look at the feedback terms yields:

$$\cos(\phi_2 + \theta_{\mathrm{H}}) \overset{(12)}{=} \omega_{\mathrm{ref}} t_{\mathrm{ext}} + \phi_{m_1} - 2\theta_{\mathrm{H}} + \theta_{\mathrm{H}} = \cos(\phi_1 - \theta_{\mathrm{H}}). \tag{28}$$

Using Equation (28) in Equation (27) yields a value of $F = 1$ which indicates that, under the assumed conditions, the intrinsic linewidth is insensitive to feedback. This result is significant as under the same condition the frequency is also feedback-insensitive, as shown in Section 2.1, regardless of lasing frequency. It is worth noting however that weak feedback is assumed in this analysis, with which the upper bound for feedback strength under which these equation are valid is not established. Nevertheless, achieving stability in the full frequency domain even under this condition is an improvement with respect to the single feedback case.

## 3. Numerical Study

Laser stability is studied by numerically evaluating the equations for the shift in lasing frequency, threshold gain and intrinsic linewidth under the revised EOF conditions, namely Equations (5) and (26), under different feedback parameters. Particular attention is given to the previously analyzed case under conditions (8), (11) and (12), which shows feedback-insensitive solutions for the lasing frequency and intrinsic linewidth. System tolerances to these conditions are explored by varying each while keeping the other two fixed. The simulated equations are plotted as a function of the unperturbed laser frequency multiplied by $t_2$, which represents the phase delay in the right external cavity for the free running laser frequency. It is kept between 0 and 1 (i.e., $\omega_{\mathrm{ref}} t_2 \in (0, 2\pi]$) given the periodicity of the numerically solved functions. Furthermore, without loss of generality, $\phi_{m_2}$ is kept fixed at zero so that the value of $\Delta\phi_m$ is selected by varying $\phi_{m_1}$. Additionally, simulations assume $\alpha_{\mathrm{H}} = 3$. This value is compatible with measurements for semiconductor lasers [48], and thus meeting condition (12) requires that $\Delta\phi_m = 2\theta_{\mathrm{H}} \simeq 2.5$. A summary of the values used in the numerical solutions is given in Table 1.

**Table 1.** Summary of values used in numerical solutions of Equations (5) and (26).

| Parameter | Value |
|---|---|
| $\omega_{\mathrm{ref}} t_{\mathrm{ext}}$ | $(0, 2\pi)$ |
| $\alpha_{\mathrm{H}}$ | 3 |
| $\kappa_2 t_2$ | 0.1581 (case 1)<br>0.3162 (case 2)<br>0.4111 (case 3) |
| $\phi_{m_2}$ | 0 |

Results are compared with the case with feedback from a single side, where:

$$\kappa_1 = 0. \tag{29}$$

In this case, as shown in [7], as feedback strength increases solutions for the lasing frequency become multi-valued. This gives rise to instabilities in the system such as mode hopping or coherence collapse regimes. The separation between single-valued and multi-valued solutions is related to the coefficient:

$$C = \gamma_{\mathrm{H}} \kappa_2 t_2, \tag{30}$$

where $C = 1$ is the critical value that separates both behaviours.

Numerical solutions of the proposed system under conditions (8) and (11) are thus compared to the single feedback case for three cases:

Case 1: $C = 0.5$. This represents the single-feedback case with a single solution, and results for various values of $\Delta\phi_m$ are shown in Figure 2. Column A, B and C show the numerical solutions for the variations in lasing frequency, threshold gain and intrinsic linewidth, given by Equations (5) and (26), respectively. By plotting the logarithm of the latter, negative values correspond to linewidth narrowing. The blue and orange plots represent the double-feedback and the single-feedback case, respectively, and these labels are maintained throughout the document.

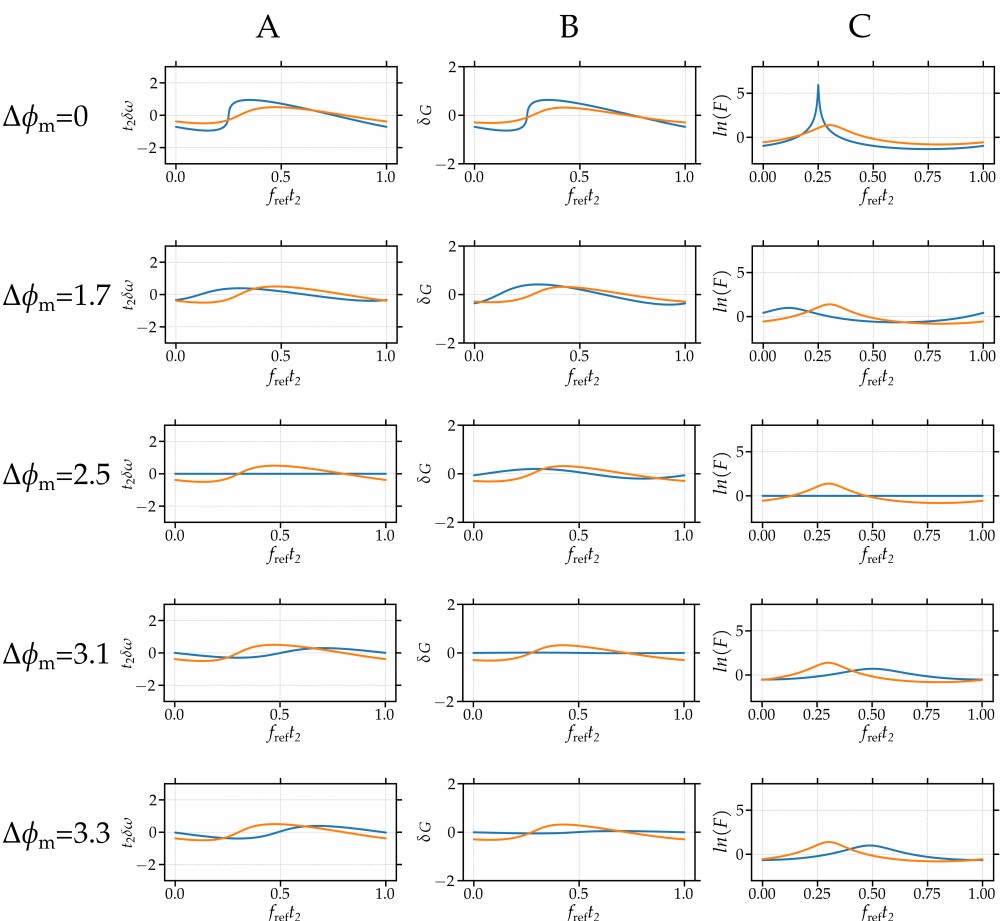

**Figure 2.** Numerical solutions for $C = 0.5$ under conditions (8) and (11) for different $\Delta\phi_m$. Double-feedback case shown in blue, single feedback case shown in orange. Column (**A**) shows the lasing frequency shift results. Column (**B**) shows the threshold gain shift. Column (**C**) shows the intrinsic linewidth variations.

The upper row shows the case where $\Delta\phi_m = 0$, i.e., additional phase shifts at, for example, the external mirrors ($\phi_{m_j}$) are kept at zero. Linewidth narrowing for a wide range of frequencies can be observed, evidenced, as mentioned, by $ln(F) < 1$. As an example, when $f_{ref}t_2 = 0.75$, a 74% and a 42% reduction in linewidth is seen with respect to the free running laser and the single feedback case, respectively. Additionally, the signal is singled valued in the full domain, yet it is close to the critical point where multi-valued solutions arise. As $\Delta\phi_m$ increases, the amplitude of the lasing frequency shift is reduced until becoming zero for all values when condition (12) is met, as expected from previous analysis. Under this condition the intrinsic linewidth does not experience fluctuations either, and the amplitude of the threshold gain fluctuations is lower than in the single feedback case, indicating better stability across the three analyzed parameters with respect to the

single feedback case. For the case of $\Delta\phi_{\rm m} = \pi$, the threshold gain shows no fluctuations as predicted by Equation (14), and while the lasing frequency and intrinsic linewidth fluctuations are no longer zero, they are less pronounced than in the single feedback case. Further increases in $\Delta\phi_{\rm m}$ show an increase in the fluctuations across all functions, and for $\phi_{\rm m} > 6$ multi-valued solutions arise.

Case 2: $C = 1$. This represents the limiting case between single and multivalued solutions in the single-feedback case. Results for various values of $\Delta\phi_{\rm m}$ are shown in Figure 3. As expected, the threshold gain is stable for $\Delta\phi_{\rm m} = \pi$, and meeting condition (12) results in a stable lasing frequency and intrinsic linewidth. As $\Delta\phi_{\rm m}$ deviates from these optimal points in either direction, the amplitude of fluctuations increase until reaching multi-valued solutions for $\Delta\phi_{\rm m} < 1.5$ and $\Delta\phi_{\rm m} > 3.5$. Comparing these results with the previous case shows that as feedback increases, the single valued solutions become more sensitive to the value of $\Delta\phi_{\rm m}$.

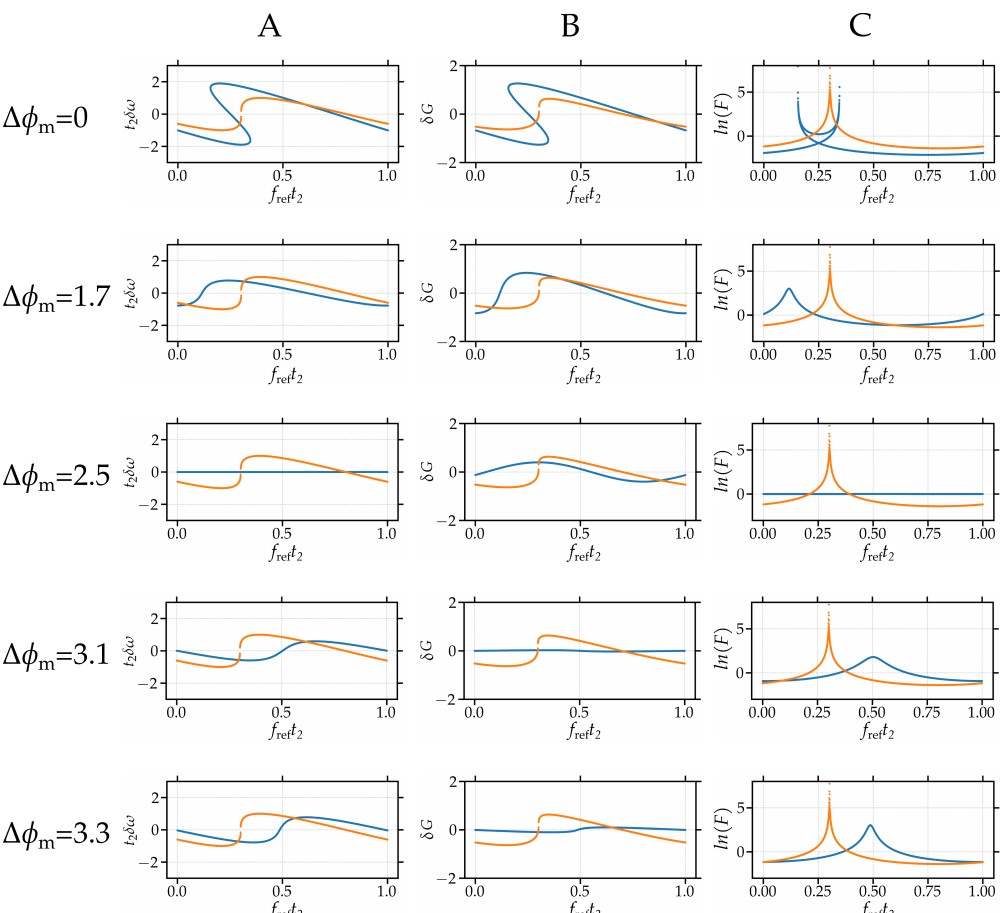

**Figure 3.** Numerical solutions for $C = 1$ under conditions (8) and (11) for different $\Delta\phi_{\rm m}$. Double-feedback case shown in blue, single feedback case shown in orange. Column (**A**) shows the lasing frequency shift results. Column (**B**) shows the threshold gain shift. Column (**C**) shows the intrinsic linewidth variations.

Case 3: $C = 1.3$. This represents the single-feedback case with multi-valued solutions. Results for various values of $\Delta\phi_{\rm m}$ are shown in Figure 4. In the single feedback case, multi-valued solutions are present in a given frequency range, and this span increases with increasing feedback strength. The multi-valued characteristics are evidenced experimentally with unstable regimes characterized by mode hopping and eventually coherence collapse for sufficiently high feedback. In contrast, the system proposed in this work shows that by tuning the value of $\Delta\phi_{\rm m}$ to meet condition (12), even with increasing feedback it is possible to achieve stable performance regardless of frequency. In the case shown in

Figure 4 for $C = 1.3$, single valued solutions can be found for $\Delta\phi_m \in (1.5, 3.3)$ which is equivalent to a phase variation of more than $90°$. Still, comparing with previous cases it is possible to see that as feedback increases, the single valued solutions tolerance with respect to $\Delta\phi_m$ decreases. Nevertheless, it is an improvement with respect to the single feedback case which shows no single value solutions across all frequencies for $C > 1$.

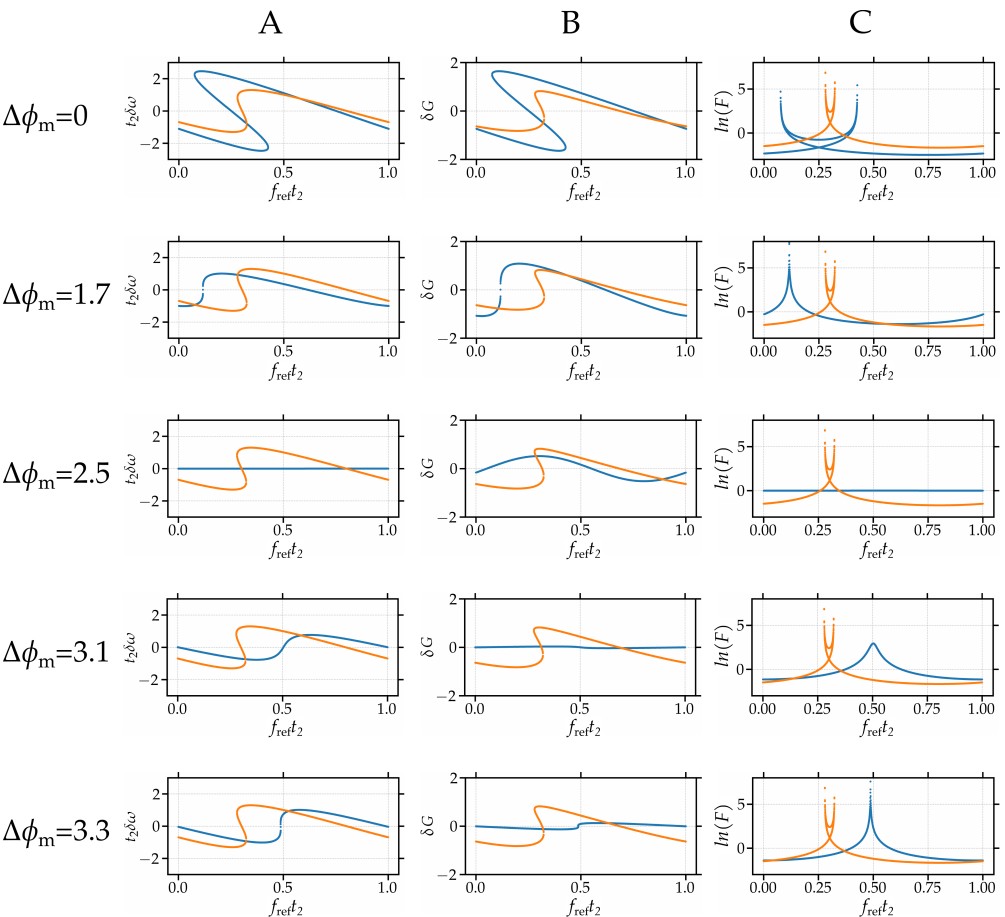

**Figure 4.** Numerical solutions for $C = 1.3$ under conditions (8) and (11) for different $\Delta\phi_m$. Double-feedback case shown in blue, single feedback case shown in orange. Column (**A**) shows the lasing frequency shift results. Column (**B**) shows the threshold gain shift. Column (**C**) shows the intrinsic linewidth variations.

Taking all cases into account, it can be seen that linewidth narrowing regions are present for all cases of $\Delta\phi_m$ analyzed. The only exception are the stable cases when meeting the three conditions (8), (11) and (12), yet the feedback-insensitivity provided by this case is also beneficial. Selecting an appropriate $\Delta\phi_m$ can thus be used to harness linewidth narrowing properties at a desired frequency.

Furthermore, system tolerances to conditions (8) and (11) are studied, while maintaining condition (12). Results for up to a 20% deviation are shown in Figures A2–A4 for cases 1, 2 and 3, respectively, where single valued solutions are observed in all cases in the full frequency span. Results show a high tolerance with respect to feedback strength. Looking at case 3, single valued solutions are obtained for $\kappa_1/\kappa_2 \in (0.2, 1.7)$. While the system is no longer feedback-insensitive, results evidence single-valued solutions that are robust with respect to condition (8). The system is more sensitive with respect to (11), with single valued solutions achieved within $t_1/t_2 \in (0.47, 1.2)$. Nevertheless, tolerances become once again stricter as feedback increases for both parameters, thus laser design is of paramount importance and should focus on meeting the discussed stability conditions. In particular, choosing equal lengths for both external cavities should suffice to meet

condition (11). Fabrication tolerances in foundry processes are the main limiting factor for time delay accuracy. To meet condition (8), a possible approach is to merge the output of the two external cavities into a single one using a coupler, which can be included in the laser on-chip. Finally, as mentioned before, condition (12) can be met by using a phase shifter, which is a mature component in active platforms.

All in all, results demonstrate that, with proper design of the laser cavity, conditions (8), (11) and (12) can be met, with which it is possible to obtain lasing frequency and intrinsic linewidth insensitivity to feedback.

## 4. Discussion

The existing mature photonic integration fabrication processes are very flexible with respect to device geometry. They are however limited by a lack of commercially available on-chip isolators, and thus new approaches are required to minimize the effect feedback has on laser stability. The current work proposes a theoretical extension of laser dynamics under EOF by considering two external reflections, one from each side of the cavity. The proposed analysis yields new laser dynamic equations. These are numerically solved, which show the existence of a stable regime with high feedback tolerances. Similar results have been seen in previous double external cavity schemes [28], with EOF on the same side of the laser cavity, which showed robust stabilization of the chaotic regime by tuning the external feedback parameters. Yet other feedback regimes are not explored, and neither analytical stability conditions nor dependence with free laser running frequency are reported. The present analysis shows that feedback-insensitivity is achievable for a wide range of feedback strengths, under conditions that can be met with current laser fabrication processes and components, such as phase shifters for meeting condition (12) and couplers for condition (8).

It has previously been shown, for the single feedback case, that tuning the feedback phase can result in linewidth narrowing [13], however this still requires low feedback levels and a precise phase shift which can suffer variations due to external parameters, such as temperature or driving currents. In the approach proposed in this work, the stability conditions do not require specific values, instead relating the feedback parameters from one side to those from the other side. For example, condition (11) only implies equal round-trip times at the external cavities, regardless of value. This allows for additional flexibility in the feedback parameters and gives versatility to the device. Furthermore, this method allows for feedback-insensitivity across the full spectra, which is not seen in the single feedback case. This is of particular importance for tunable lasers, as all lasing frequencies are thus equally affected. Additionally, it relaxes the need for an isolator, reducing the cost and size of packaging processes. Another significant improvement of the proposed method with respect to the single feedback case is the increase in feedback tolerance: by choosing feedback parameters close to the stability conditions of Equations (11) and (12), higher levels of feedback strength are allowed without seeing multi-valued solutions, which results in experimentally seen mode-hopping. As a weak feedback approximation is used, the upper bound for feedback tolerance cannot be extracted from this analysis. Despite this, even under this approximation, tolerances are higher than that of the single feedback case. Furthermore, this system has a high tolerance to deviations from the optimal stability conditions as analyzed in the previous section. Linewidth narrowing can be achieved in these cases for certain frequency values, which can be tuned by selecting appropriate feedback parameters, as was the case for single feedback conditions, while maintaining stable solutions.

Finally, while the dynamics under consideration are complex, the laser system itself involves a straight-forward configuration using widely used on-chip components, which are available in generic foundry platforms. Previously studied methods to reduce feedback sensitivity include resourceful yet intricate designs. The proposed system is, in contrast, potentially easier to design, fabricate and characterize. An experimental study of this laser system is essential to validate the obtained results, and more importantly to explore the

limitations of the model, and is the next step for a more comprehensive understanding of the proposed system. It is worth noting that, for the single feedback case, the Lang-Kobayashi approach yields results compatible with experimental data [49–51], which suggests that future experimental realization of the method here presented will be compatible with its theoretical predictions.

## 5. Conclusions

This work explores an extension of the theoretical background of EOF. By assuming that feedback couples into the laser cavity from both sides, new dynamic equations are found for the lasing frequency, the threshold gain and the intrinsic linewidth. These are numerically evaluated to analyze laser stability. Solutions with linewidth reduction are observed, where tuning of the feedback parameters yielded a 74% and a 42% reduction with respect of the free running laser and the single feedback case, respectively. Results also show the existence of a stable solution, with feedback-insensitive lasing frequency and intrinsic linewidth, regardless of the lasing frequency. This case is obtained by tuning the phase of the feedback field, for external cavities with equal lengths and coupling factors. Furthermore, the feedback-insensitive case exists regardless of the feedback strength, within a weak feedback approximation, which is an major improvement with respect to the single feedback case. Additionally, the stability conditions show good tolerances with respect to all feedback parameters, albeit they become stricter as the feedback strenght increases. Choosing feedback parameters close to the feedback-insensitive conditions ensures stable solutions that are feedback tolerant. Finally, the proposed system relies on few components in straight-forward configurations, and the stable conditions can be met with mature components available in generic foundry platforms. This enables close to market, low cost, feedback-tolerant semiconductor lasers which have direct applications in multiple fields that rely on stable laser sources, such as coherent communications and spectroscopy.

**Author Contributions:** Formal analysis, M.F.B.; Funding acquisition, N.V.; Project administration, N.V.; Resources, N.V.; Supervision, N.V.; Writing—original draft, M.F.B.; Writing—review & editing, M.F.B. and N.V. All authors have read and agreed to the published version of the manuscript.

**Funding:** We acknowledge support from Independent Research Fund Denmark.

**Institutional Review Board Statement:** Not Applicable.

**Informed Consent Statement:** Not Applicable.

**Data Availability Statement:** Not Applicable.

**Conflicts of Interest:** The authors declare no conflict of interest.

## Appendix A. Amplitude and Phase Conditions

To obtain the revised lasing conditions resulting from the additional feedback component, extracting the amplitude and phase of the effective reflection coefficients is needed. In polar notation:

$$\rho_j^{\text{eff}} = \left|\rho_j^{\text{eff}}\right| e^{i\varphi_j}, \tag{A1}$$

where the magnitude is computed as:

$$\left|\rho_j^{\text{eff}}\right|^2 \bigg/ \rho_j^2 \stackrel{(3)}{=} \left[1 + \kappa_j t_{\text{cav}} \cos(\phi_j)\right]^2 + \left[\kappa_j t_{\text{cav}} \sin(\phi_j)\right]^2$$

$$= 1 + 2\kappa_j t_{\text{cav}} \cos(\phi_j) + \kappa_j^2 t_{\text{cav}}^2. \tag{A2}$$

Assuming condition (1) of weak external feedback, the last term on the right-hand side of Equation (A2) is neglected, resulting in:

$$\left| \rho_j^{\text{eff}}(\omega_{\text{FB}}) \right| \Big/ \rho_i = \sqrt{1 + 2\kappa_j t_{\text{cav}} \cos(\phi_j)} \overset{(1)}{\approx} 1 + \kappa_j t_{\text{cav}} \cos(\phi_j). \tag{A3}$$

The phase of the effective reflection coefficient is extracted from:

$$\varphi_j = \arctan\left[ \left( \rho_j^{\text{eff}} \right)'' \Big/ \left( \rho_j^{\text{eff}} \right)' \right] \overset{(3)}{=} \arctan\left[ \frac{\pm \kappa_j t_{\text{cav}} \sin(\phi_j)}{1 + \kappa_j t_{\text{cav}} \cos(\phi_j)} \right]$$

$$\overset{(1)}{\approx} \arctan[\pm \kappa_j t_{\text{cav}} \sin(\phi_j)] \overset{(1)}{\approx} \pm \kappa_j t_{\text{cav}} \sin(\phi_j). \tag{A4}$$

The fields traveling forward and backward in the laser cavity, $\mathcal{E}_f$ and $\mathcal{E}_b$ shown in Figure A1, can now be related by the effective reflection coefficients:

$$\mathcal{E}_f(z = 0) = \rho_1^{\text{eff}} \mathcal{E}_b(z = 0) \tag{A5a}$$

$$\mathcal{E}_b(z = L) = \rho_2^{\text{eff}} \mathcal{E}_f(z = L). \tag{A5b}$$

Using the propagation constant:

$$\beta \equiv n\omega/c \,, \tag{A6}$$

with $n$ the effective refractive index of the lasing mode and $c$ the speed of light in vacuum, the fields can be written as:

$$\mathcal{E}_f = A_f e^{-i\beta z + \frac{1}{2}(\Gamma g - \alpha)z} \tag{A7a}$$

$$\mathcal{E}_b = A_b e^{-i\beta(L-z) + \frac{1}{2}(\Gamma g - \alpha)(L-z)}, \tag{A7b}$$

where $g$ is the gain coefficient and $\alpha$ is the attenuation coefficient. Replacing Equation (A5) into Equation (A7):

$$\mathcal{E}_{f0} \overset{(A1)}{=} \left| \rho_2^{\text{eff}} \right| e^{i\varphi_2} A_b e^{-i\beta L + (\Gamma g - \alpha)L/2} \tag{A8a}$$

$$\mathcal{E}_{b0} \overset{(A1)}{=} \left| \rho_1^{\text{eff}} \right| e^{i\varphi_1} A_f e^{-i\beta L + (\Gamma g - \alpha)L/2}, \tag{A8b}$$

and inserting Equation (A8a) into Equation (A8b) results in:

$$1 = \left| \rho_1^{\text{eff}} \right| e^{i\varphi_1} \left| \rho_2^{\text{eff}} \right| e^{i\varphi_2} e^{-2i\beta L + (\Gamma g - \alpha)L} = \left| \rho_1^{\text{eff}} \rho_2^{\text{eff}} \right| e^{-i(2\beta L - \varphi_1 - \varphi_2)} e^{(\Gamma g - \alpha)L}. \tag{A9}$$

Once lasing has been established, the gain assumes its threshold value:

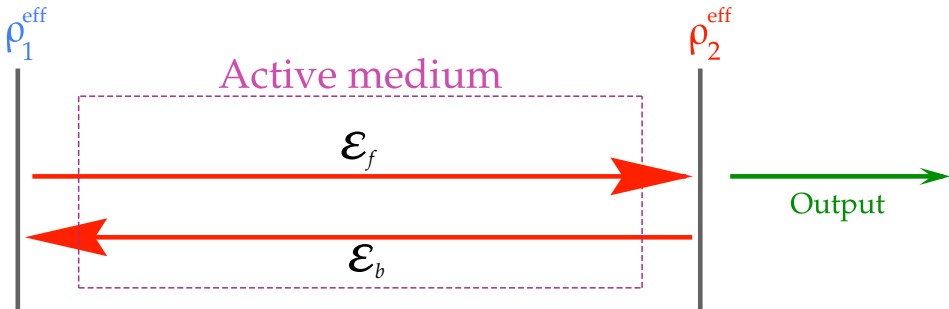

**Figure A1.** Schematic of the effective cavity of the laser, resulting from calculating effective reflection coeffitients.

$$g = g_{\text{FB}}, \tag{A10}$$

where $g_{\mathrm{FB}}$ is the threshold gain with feedback. Thus, Equation (A9) yields a lasing condition for the amplitude:

$$1 = \left| \rho_1^{\mathrm{eff}} \rho_2^{\mathrm{eff}} \right| e^{(\Gamma g_{\mathrm{FB}} - \alpha) L}$$

$$\stackrel{\text{(A10)(A3)}}{=\approx} \rho_2 \rho_1 [1 + \kappa_2 t_{\mathrm{cav}} \cos(\phi_2)][1 + \kappa_1 t_{\mathrm{cav}} \cos(\phi_1)] e^{(\Gamma g_{\mathrm{FB}} - \alpha) L}, \tag{A11}$$

and the phase:

$$2\pi m = 2\beta L - \varphi_2 - \varphi_1 \stackrel{\text{(A4)}}{=} 2\beta L + \kappa_2 t_{\mathrm{cav}} \sin(\phi_2) - \kappa_1 t_{\mathrm{cav}} \sin(\phi_1) \quad , \quad m \in \mathbb{Z}. \tag{A12}$$

where the influence of feedback gives rise to two terms in Equations (A11) and (A12), one from each side. The new lasing conditions result in a variation of the lasing frequency and threshold gain of the system, and thus have to be studied to determine the laser dynamics.

**Appendix B. Threshold Gain Reduction and Lasing Frequency Shift**

Under feedback from both sides of the laser cavity, new lasing conditions are found which subsequently result in a shift of the laser threshold gain and lasing frequency with respect to the case without feedback, in which:

$$\kappa_j = 0. \tag{A13}$$

In this case, the amplitude condition from Equation (A11) becomes:

$$1 \stackrel{\text{(A13)}}{=} \rho_2 \rho_1 e^{(\Gamma g_{\mathrm{th}} - \alpha) L}. \tag{A14}$$

Using the expansion:

$$\ln(1 + x) \simeq x, \tag{A15}$$

the threshold gain reduction due to feedback can be found by computing the ratio between Equations (A11) and (A14):

$$1 = \frac{\rho_2 \rho_1 [1 + \kappa_2 t_{\mathrm{cav}} \cos(\phi_2)][1 + \kappa_1 t_{\mathrm{cav}} \cos(\phi_1)] e^{(\Gamma g_{\mathrm{FB}} - \alpha) L}}{\rho_2 \rho_1 e^{\Gamma g_{\mathrm{ref}} - \alpha L}}$$

$$= [1 + \kappa_2 t_{\mathrm{cav}} \cos(\phi_2)][1 + \kappa_1 t_{\mathrm{cav}} \cos(\phi_1)] e^{\Gamma (g_{\mathrm{FB}} - g_{\mathrm{ref}}) L}$$

$$\stackrel{\text{(2)}}{\Leftrightarrow} G_{\mathrm{FB}} - G_{\mathrm{ref}} \approx -2\kappa_2 \cos(\phi_2) - 2\kappa_1 \cos(\phi_1). \tag{A16}$$

The relation between the right hand terms determines the threshold gain reduction, as discussed in Section 2.1.

The phase lasing condition from Equation (A12) yields the lasing frequency shift equation. Consider the following definitions related to the effective refractive index [52]:

$$n \equiv n' + in'' \tag{A17a}$$

$$n_{\mathrm{g}} \equiv n + \omega \frac{\partial n}{\partial \omega} \tag{A17b}$$

$$n'' \equiv -\frac{cG}{2\omega v_{\mathrm{g}}}, \tag{A17c}$$

$$\alpha_{\text{H}} \equiv \Delta n' / \Delta n'' \tag{A17d}$$

$$\frac{\partial n}{\partial N} = \frac{\partial n}{\partial n''} \frac{\partial n''}{\partial N}$$

$$\stackrel{\text{(A17a)(A17c)(A17d)}}{=} -\frac{\partial G}{\partial N} \frac{\alpha_{\text{H}} c}{2\omega v_{\text{g}}}, \tag{A17e}$$

where $n_{\text{g}}$ is the group refractive index. To find the lasing frequency shift equation, calculating the change in $\beta$ is first needed:

$$c\delta\beta \stackrel{\text{(A6)}}{=} \delta(n\omega) = \omega\delta n + n\delta\omega \stackrel{\text{(2c)}}{=} \omega\left[\frac{\partial n}{\partial N}(N - N_{\text{th}}) + \frac{\partial n}{\partial \omega}\delta\omega\right] + n\delta\omega$$

$$\stackrel{\text{(A17b)(A17e)}}{=} -\frac{G_{\text{FB}} - G_{\text{th}}}{2v_{\text{g}}}\alpha_{\text{H}} c + n_{\text{g}}\delta\omega \stackrel{\text{(5a)}}{=} [\kappa_2 \cos(\phi_2) + \kappa_1 \cos(\phi_1)]\frac{\alpha_{\text{H}} c}{v_{\text{g}}} + n_{\text{g}}\delta\omega, \tag{A18}$$

with which:

$$2L\delta\beta \stackrel{\text{(5a)}}{=} [\kappa_2 \cos(\phi_2) + \kappa_1 \cos(\phi_1)]\alpha_{\text{H}} t_{\text{cav}} + t_{\text{cav}}\delta\omega. \tag{A19}$$

Furthermore, using:

$$\sin[\arctan(x)] = \frac{x}{\sqrt{1 + x^2}} \tag{A20a}$$

$$\cos[\arctan(x)] = \frac{1}{\sqrt{1 + x^2}} \tag{A20b}$$

$$\sin(x \pm y) = \sin(x)\cos(y) \pm \cos(x)\sin(y) \tag{A20c}$$

$$\cos(\theta_{\text{H}}) \stackrel{\text{(4e)(A20b)(4d)}}{=} 1/\gamma_{\text{H}}, \tag{A20d}$$

the following can be computed:

$$\alpha_{\text{H}} \cos(\phi_j) \stackrel{\text{(A20a)(4d)(4e)}}{=} \gamma_{\text{H}} \sin(\theta_{\text{H}}) \cos(\phi_j)$$

$$\stackrel{\text{(A20c)}}{=} \gamma_{\text{H}} \left[\sin(\theta_{\text{H}} \pm \phi_j) \mp \cos(\theta_{\text{H}}) \sin(\phi_j)\right] \tag{A21}$$

$$\stackrel{\text{(A20d)}}{=} \gamma_{\text{H}} \left[\sin(\theta_{\text{H}} \pm \phi_j) \mp \sin(\phi_j)/\gamma_{\text{H}}\right]$$

$$\Leftrightarrow \alpha_{\text{H}} \cos(\phi_j) \pm \sin(\phi_j) = \gamma_{\text{H}} \sin(\theta_{\text{H}} \pm \phi_j).$$

Finally, using the phase condition in Equation (A12), and assuming without generality loss that $m = 0$, it is possible to compute:

$$2\pi m \stackrel{\text{(A19)}}{=} t_{\text{cav}}\delta\omega + \kappa_2 t_{\text{cav}}[\alpha_{\text{H}} \cos(\phi_2) + \sin(\phi_2)] + \kappa_1 t_{\text{cav}}[\alpha_{\text{H}} \cos(\phi_1) - \sin(\phi_1)]$$

$$\stackrel{\text{m=0}}{\Leftrightarrow} \delta\omega = -\kappa_2[\alpha_{\text{H}} \cos(\phi_2) + \sin(\phi_2)] + \kappa_1[\alpha_{\text{H}} \cos(\phi_1) - \sin(\phi_1)] \tag{A22}$$

$$\delta\omega \stackrel{\text{(A21)}}{=} -\gamma_{\text{H}}[\kappa_2 \sin(\phi_2 + \theta_{\text{H}}) + \kappa_1 \sin(\phi_1 - \theta_{\text{H}})],$$

which describes the lasing frequency shift as a function of the feedback parameters $\kappa_1, \kappa_2, \phi_1$ and $\phi_2$, Similarly to the threshold gain reduction, the interaction between the two right hand terms determines the lasing frequency stability which is discussed in Section 2.1.

**Appendix C. Deriving the Rate Equation for the Intensity and Phase**

To further inspect the laser dynamics, the rate equations for the intensity and phase must be studied. Considering Equation (15), and a slowly varying electric field given by:

$$\mathcal{E}(t) = \mathcal{A}(t)e^{-i\omega_{\mathrm{FB}}t}, \tag{A23}$$

and following the approach from Lang and Kobayashi [39], the laser field equation which considers EOF from both sides of the laser cavity can be written as:

$$\dot{\mathcal{E}} = \left(-i\omega_{\mathrm{ref}} + \Delta G\frac{1 - i\alpha_{\mathrm{H}}}{2}\right)\mathcal{E}(t) + \kappa_2\mathcal{E}(t - t_2) + \kappa_1\mathcal{E}(t + t_1)$$

$$\overset{(A23)}{\Longleftrightarrow} \frac{d\left[\mathcal{A}(t)e^{-i\omega_{\mathrm{FB}}t}\right]}{dt} \overset{(15)}{=} e^{-i\omega_{\mathrm{FB}}t}\left[\left(-i\omega_{\mathrm{ref}} + \Delta G\frac{1 - i\alpha_{\mathrm{H}}}{2}\right)\mathcal{A}(t) + \kappa_2\mathcal{A}(t - t_2)e^{i\phi_2} + \kappa_1\mathcal{A}(t + t_2)e^{-i\phi_1}\right] \tag{A24}$$

$$\Longleftrightarrow \dot{\mathcal{A}}(t) \overset{(2c)(2b)}{=} \left(i\delta\omega + \Delta G\frac{1 - i\alpha_{\mathrm{H}}}{2}\right)\mathcal{A}(t) + \kappa_2\mathcal{A}(t - t_2)e^{i\phi_2} + \kappa_1\mathcal{A}(t + t_2)e^{-i\phi_1}.$$

The last two right-hand terms appear as a result of the imposed feedback conditions, each term to account for feedback on each side of the cavity. In the case without feedback the lasing frequency become $\omega_{\mathrm{FB}} = \omega_{\mathrm{ref}}$ and $\kappa_j = 0$, thus recovering the no-feedback field Equation [42]. The slowly varying field amplitude $\mathcal{A}$ can be modeled as Equation (15a), and thus the rate equations for the photon number $S$ and phase $\phi$ can be found using:

$$\dot{S} = \frac{d[\mathcal{A}\mathcal{A}^*]}{dt} = \mathcal{A}\dot{\mathcal{A}}^* + \mathcal{A}^*\dot{\mathcal{A}} \tag{A25a}$$

$$\dot{\phi} = -\frac{1}{S}\Im(\mathcal{A}^*\dot{\mathcal{A}}). \tag{A25b}$$

Replacing Equations (15a) and (A24) into Equation (A25a) the photon rate equation reads:

$$\dot{S} \overset{(15d)(15e)}{=} S\Delta G + 2\mathcal{S}_2^-\sqrt{S}\cos(\Delta\Phi_2^-) + 2\mathcal{S}_1^+\sqrt{S}\cos(\Delta\Phi_1^+). \tag{A26}$$

In the case of the phase, its rate equation comes from replacing Equations (15a) and (A24) into Equation (A25b):

$$\dot{\phi} \overset{(15)}{=} \frac{\Delta G}{2}\alpha_{\mathrm{H}} - \delta\omega - \frac{\mathcal{S}_2^-}{\sqrt{S}}\sin(\Delta\Phi_2^-) - \frac{\mathcal{S}_1^+}{\sqrt{S}}\sin(\Delta\Phi_1^+). \tag{A27}$$

Equations (A26) and (A27) are the amplitude and phase rate equations for the laser system proposed in this work. These are the starting point to compute the frequency noise PSD, and extract the intrinsic linewidth.

**Appendix D. Small-Signal Analysis**

To find the FN PSD, the system shown in Equation (16) is to be solved. This is done using the small-signal analysis proposed in Equation (19). Assuming a narrow-linewidth laser, i.e., a long coherence time with respect to the external cavity lenghts:

$$t_{\mathrm{ext}} < t_{\mathrm{coh}}, \tag{A28}$$

the following approximation is valid:

$$\Omega t_{\mathrm{ext}} \ll 1. \tag{A29}$$

By linearizing the following expressions:

$$\sqrt{\frac{S(t \pm t_j)}{S(t)}} \overset{(A28)}{\approx} = \sqrt{1 \pm \frac{\dot{S}}{S} t_j} \overset{(19a)}{\approx} \sqrt{1 \pm \frac{i\Omega' S_\Delta}{S_0} t_j} \overset{(A29)}{\approx} 1 \pm \frac{i\Omega' S_\Delta}{2S_0} t_j \tag{A30a}$$

$$\phi - \phi(t \pm t_j) \overset{(A28)}{\approx} \phi - \phi \mp t_j \dot{\phi} \overset{(19b)}{=} \mp i t_j \Omega' \phi_\Delta , \tag{A30b}$$

it is possible to rewrite Equation (16) as:

$$i\Omega' S_\Delta \overset{(A30)}{=} S\Delta G + R_{sp} + 2\frac{\kappa_2}{t_{cav}} S\left(1 - \frac{i\Omega' S_\Delta}{2S_0} t_2\right) \cos\left(i\Omega' \phi_\Delta t_2 + \phi_2\right)$$

$$+ 2\kappa_1 S\left(1 + \frac{i\Omega' S_\Delta}{2S_0} t_1\right) \cos\left(i\Omega' \phi_\Delta t_1 + \phi_1\right) + F_S$$

$$i\Omega' \phi_\Delta \overset{(A30)}{=} \alpha_H \frac{\Delta G}{2} - \delta\omega - \frac{\kappa_2}{t_{cav}} \left(1 - \frac{i\Omega' S_\Delta}{2S_0} t_2\right) \sin\left(i\Omega' \phi_\Delta t_2 + \phi_2\right) \tag{A31a}$$

$$+ \kappa_1 \left(1 + \frac{i\Omega' S_\Delta}{2S_0} t_1\right) \sin\left(i\Omega' \phi_\Delta t_1 + \phi_1\right) + F_\phi \tag{A31b}$$

$$i\Omega' N_\Delta \overset{(19)(A30)}{=} I - G_{FB} S - N\tau_{sp}^{-1} + F_N. \tag{A31c}$$

Solving Equation (A31) requires the steady-state solution of Equation (16). Under stationary conditions:

$$\dot{S} = 0 \Rightarrow S(t) = S(t \pm t_j) = S_0 \tag{A32a}$$

$$\dot{\phi} = 0 \Rightarrow \phi(t) = \phi(t \pm t_j) \tag{A32b}$$

$$\dot{N} = 0 \Rightarrow N(t) = N(t \pm t_j) = N_0, \tag{A32c}$$

the steady-state equations are:

$$\tau_{ph}^{-1} = G_0 + 2\kappa_2 \cos(\phi_2) + 2\kappa_1 \cos(\phi_1) + \frac{R_{sp}}{S_0} \tag{A33a}$$

$$\delta\omega = \alpha_H \frac{G_0 - \tau_{ph}^{-1}}{2} - \kappa_2 \sin(\phi_2) + \kappa_1 \sin(\phi_1) \tag{A33b}$$

$$I = G_0 S_0 + N_0 \tau_{sp}^{-1}, \tag{A33c}$$

where the Langevin noise terms are not included as their mean value is zero. Next, using Equation (A33) and the following expansions:

$$\sin(x + \Delta) \approx \sin(x) + \Delta \cos(x) \tag{A34a}$$

$$\cos(x + \Delta) \approx \cos(x) - \Delta \sin(x), \tag{A34b}$$

Equation (A31a) can be rewritten as:

$$i\Omega' S_\Delta \overset{(19)(A33)(20)}{\approx} \left(G_0 + a_g N_\Delta - \left\{G_0 + 2[\kappa_2 \cos(\phi_2) + \kappa_1 \cos(\phi_1)] + \frac{R_{sp}}{S_0}\right\}\right)(S_0 + S_\Delta)$$

$$+ 2\Bigg\{ \kappa_2(S_0 + S_\Delta)\left[\cos(\phi_2) - i\Omega' \phi_\Delta t_2 \sin(\phi_2)\right]$$

$$- \kappa_2(S_0 + S_\Delta)\frac{i\Omega' S_\Delta}{2S_0} t_2 \left[\cos(\phi_2) - i\Omega' \phi_\Delta t_2 \sin(\phi_2)\right]$$

$$+ \kappa_1(S_0 + S_\Delta)\left[\cos(\phi_1) - i\Omega' \phi_\Delta t_1 \sin(\phi_1)\right]$$

$$+ \kappa_1(S_0 + S_\Delta)\frac{i\Omega' S_\Delta}{2S_0} t_1 \left[\cos(\phi_1) - i\Omega' \phi_\Delta t_1 \sin(\phi_1)\right] \Bigg\} + R_{sp} + F_S.$$

Simplifying this equation, and neglecting the quadratic terms yields:

$$i\Omega' S_\Delta \overset{(20)}{\approx} a_g N_\Delta S_0 - \zeta_s S_\Delta - i\Omega' \phi_\Delta 2\mathsf{K}_s S_0 + i\Omega' S_\Delta (\kappa_1^c - \kappa_2^c) + F_S. \tag{A35a}$$

In a similar way, Equation (A31b) can be rewritten using Equations (A33) and (20):

$$i\Omega' \phi_\Delta \approx \frac{\alpha_H}{2}\left(G_0 + a_g N_\Delta - \tau_{ph}^{-1}\right) - \left[\alpha_H \frac{G_0 - \tau_{ph}^{-1}}{2} - \kappa_2 \sin(\phi_2) + \kappa_1 \sin(\phi_1)\right]$$

$$- \kappa_2\left(1 - \frac{i\Omega' S_\Delta}{2S_0}t_2\right)\left[\sin(\phi_2) + i\Omega' \phi_\Delta t_2 \cos(\phi_2)\right] \tag{A35b}$$

$$+ \kappa_1\left(1 + \frac{i\Omega' S_\Delta}{2S_0}t_1\right)\left[\sin(\phi_1) + i\Omega' \phi_\Delta t_1 \cos(\phi_1)\right] + F_\phi$$

$$\Leftrightarrow 2i\Omega' \phi_\Delta \overset{(20a)(20b)}{=} \alpha_H a_g N_\Delta + 2(\kappa_1^c - \kappa_2^c)i\Omega' \phi_\Delta + \frac{i\Omega' S_\Delta}{S_0}\mathsf{K}_s + 2F_\phi.$$

Finally, Equation (A31c) can be rewritten as:

$$i\Omega' N_\Delta \overset{(19)(A33)}{=} \overset{(20)}{\approx} G_0 S_0 - (G_0 + a_g N_\Delta)S_0 - (G_0 + a_g N_\Delta)S_\Delta - \tau_{sp}^{-1} N_\Delta + F_N$$

$$\Leftrightarrow i\Omega' N_\Delta \overset{(20h)}{=} -\tau_e^{-1} N_\Delta - G_0 S_\Delta + F_N. \tag{A35c}$$

The linearized rate equations of the laser under study are thus Equations (A35a)–(A35c), from which the power spectral density, and subsequently the intrinsic linewidth, can be computed.

**Appendix E. Power Spectral Density**

The next step is to find an expression for $\phi_{0p}$ from which the FN PSD, and thus the laser intrinsic linewidth, can be computed. The following definitions are convenient:

$$A_\phi \equiv \left(i\Omega + \tau_e^{-1}\right)\left(i\Omega \mathsf{K}_c + \frac{a_g G_0}{i\Omega + \tau_e^{-1}}S_0 + \zeta_s\right) \tag{A36a}$$

$$2A_S \equiv i\Omega \mathsf{K}_s \frac{i\Omega + \tau_e^{-1}}{S_0} - \alpha_H a_g G_0 \tag{A36b}$$

$$2A_N \equiv \frac{\alpha_H A_\phi + 2S_0 A_S}{i\Omega + \tau_e^{-1}}a_g \tag{A36c}$$

$$B_\phi \equiv \mathsf{K}_c A_\phi - \alpha_H a_g G_0 S_0 \mathsf{K}_s + i\Omega \mathsf{K}_s^2\left(i\Omega + \tau_e^{-1}\right). \tag{A36d}$$

First, $N_{0p}$ is extracted from Equation (21c):

$$N_{0p} = \frac{\widehat{F}_N - G_0 S_{0p}}{i\Omega + \tau_e^{-1}}. \tag{A37}$$

Next, replacing Equation (A37) into Equation (21a) yields the expression for $S_{0\mathrm{p}}$:

$$(i\Omega\mathsf{K}_c + \zeta_S)S_{0\mathrm{p}} = a_g \frac{\widehat{F}_N - G_0 S_{0\mathrm{p}}}{i\Omega + \tau_\mathrm{e}^{-1}} S_0 - 2i\Omega S_0 \mathsf{K}_s \phi_{0\mathrm{p}} + \widehat{F}_S$$

$$\Leftrightarrow \frac{A_\phi}{i\Omega + \tau_\mathrm{e}^{-1}} S_{0\mathrm{p}} = a_g S_0 \frac{\widehat{F}_N}{i\Omega + \tau_\mathrm{e}^{-1}} - 2i\Omega S_0 \mathsf{K}_s \phi_{0\mathrm{p}} + \widehat{F}_S \qquad (A38)$$

$$\Leftrightarrow S_{0\mathrm{p}} \overset{(A36a)}{=} \frac{a_g S_0 \widehat{F}_N - 2iS_0\Omega\mathsf{K}_s\left(i\Omega + \tau_\mathrm{e}^{-1}\right)\phi_{0\mathrm{p}} + \left(i\Omega + \tau_\mathrm{e}^{-1}\right)\widehat{F}_S}{A_\phi}.$$

Finally, inserting Equation (A37) and (A38) into Equation (21b) and grouping the terms with $\phi_{0\mathrm{p}}$, $\widehat{F}_S$, $\widehat{F}_\phi$ and $\widehat{F}_N$ yields:

$$2i\Omega B_\phi\,\phi_{0\mathrm{p}} \overset{(A36)}{=} \left[-\alpha_\mathrm{H}a_g G_0 S_0 + \alpha_\mathrm{H} A_\phi + i\Omega\mathsf{K}_s\left(i\Omega + \tau_\mathrm{e}^{-1}\right)\right]\frac{a_g}{i\Omega + \tau_\mathrm{e}^{-1}}\,\widehat{F}_N + 2A_S\,\widehat{F}_S + 2\,A_\phi\widehat{F}_\phi$$

$$\overset{(A36b)}{=} \left(\alpha_\mathrm{H} A_\phi + 2S_0 A_S\right)\frac{a_g}{i\Omega + \tau_\mathrm{e}^{-1}}\,\widehat{F}_N + A_S\,\widehat{F}_S + 2A_\phi\widehat{F}_\phi \qquad (A39)$$

$$\Leftrightarrow \phi_{0\mathrm{p}} \overset{(A36c)}{=} \frac{A_N\,\widehat{F}_N + A_S\,\widehat{F}_S + A_\phi\widehat{F}_\phi}{i\Omega B_\phi}.$$

With Equation (A39) it is possible to calculate an expression for Equation (22):

$$2\pi^2|B_\phi|^2 S_f^{(1)} \overset{(22)}{=} \Omega^2|B_\phi|^2\left\langle \phi_{0\mathrm{p}}(\Omega)\phi_{0\mathrm{p}}^*(\Omega)\right\rangle$$

$$\overset{(A39)}{=} \Omega^2|B_\phi|^2\left\langle \frac{A_N\,\widehat{F}_N + A_S\,\widehat{F}_S + A_\phi\widehat{F}_\phi}{i\Omega B_\phi}\left(\frac{A_N\,\widehat{F}_N + A_S\,\widehat{F}_S + A_\phi\widehat{F}_\phi}{i\Omega B_\phi}\right)^*\right\rangle \qquad (A40)$$

$$= \left\langle \left(A_N\,\widehat{F}_N + A_S\,\widehat{F}_S + A_\phi\widehat{F}_\phi\right)\left(A_N\,\widehat{F}_N + A_S\,\widehat{F}_S + A_\phi\widehat{F}_\phi\right)^*\right\rangle.$$

It can be seen from Equation (A36) that the coefficients $A_i$ are independent of time, and assuming an ergodic process they can be taken out of the average in Equation (A40), obtaining:

$$\pi^2|B_\phi|^2 S_f^{(1)} \overset{(17b)}{=} |A_N|^2 D_{NN} + |A_S|^2 D_{SS} + |A_\phi|^2 D_{\phi\phi} + \left(A_S A_N^* + A_N A_S^*\right) D_{SN} \qquad (A41)$$

$$\Leftrightarrow 4\pi^2|B_\phi|^2 S_f^{(1)} \overset{(A36)(23)}{=} \left[\left(\zeta_S\alpha_\mathrm{H}a_g\right)^2 + \Omega^2 a_g^2\mathsf{F}_2^2\right] D_{NN}$$

$$+ \left[\Omega^4\mathsf{K}_c^2 + \Omega^2\mathsf{F}_0 + \left(\tau_\mathrm{e}^{-1}\zeta_S + a_g G_0 S_0\right)^2\right] 4D_{\phi\phi}$$

$$+ \left\{\Omega^2\left[\left(\tau_\mathrm{e}^{-1}\frac{\mathsf{K}_s}{S_0}\right)^2 + 2\alpha_\mathrm{H}a_g G_0\frac{\mathsf{K}_s}{S_0}\right] + \Omega^4\left(\frac{\mathsf{K}_s}{S_0}\right)^2 + \left[\alpha_\mathrm{H}a_g G_0\right]^2\right\} D_{SS}$$

$$- 2\left[-\alpha_\mathrm{H}^2 a_g^2 G_0\zeta_S + \Omega^2\frac{\mathsf{K}_s}{S_0}a_g\left(\tau_\mathrm{e}^{-1}\mathsf{F}_2 - \zeta_S\alpha_\mathrm{H}\right)\right] D_{SS}$$

$$\Leftrightarrow 2\pi^2 S_f^{(1)} \overset{(23)}{=} \frac{\Lambda_4\Omega^4 + \Lambda_2\Omega^2 + \Lambda_0}{2|B_\phi|^2}. \qquad (A42)$$

Furthermore, using the following definitions:

$$\Delta_4 \equiv \mathsf{F}_1^2 \tag{A43a}$$

$$\Delta_2 \equiv \mathsf{K}_c^2 \zeta_S^2 + \tau_e^{-2} \mathsf{F}_1^2 - 2\mathsf{F}_1 a_g G_0 S_0 \mathsf{F}_3 \tag{A43b}$$

$$\Delta_0 \equiv \left( a_g G_0 S_0 \mathsf{F}_3 + \mathsf{K}_c \zeta_S \tau_e^{-1} \right)^2, \tag{A43c}$$

the expression for $B_\phi$ from Equation (A36d) can be rewritten as:

$$B_\phi = a_g G_0 S_0 \mathsf{F}_3 - \Omega^2 \mathsf{F}_1 + i\Omega \left( \mathsf{K}_c \zeta_S + \tau_e^{-1} \mathsf{F}_1 \right) + \mathsf{K}_c \zeta_S \tau_e^{-1} \tag{A44a}$$

$$\Leftrightarrow |B_\phi|^2 = \Omega^2 \left( \mathsf{K}_c \zeta_S + \tau_e^{-1} \mathsf{F}_1 \right)^2 + \left( a_g G_0 S_0 \mathsf{F}_3 - \Omega^2 \mathsf{F}_1 + \mathsf{K}_c \zeta_S \tau_e^{-1} \right)^2$$

$$\overset{(A43c)}{=} \Omega^4 \Delta_4 + \Omega^2 \left[ \left( \mathsf{K}_c \zeta_S + \tau_e^{-1} \mathsf{F}_1 \right)^2 - 2\mathsf{F}_1 a_g G_0 S_0 \mathsf{F}_3 - 2\mathsf{F}_1 \mathsf{K}_c \zeta_S \tau_e^{-1} \right] + \Delta_0$$

$$\overset{(A43c)}{=} \Omega^4 \Delta_4 + \Omega^2 \Delta_2 + \Delta_0, \tag{A44b}$$

with which the expression of the FN PSD becomes:

$$4\pi^2 S_f^{(1)} \overset{(23)}{=} \frac{\Lambda_4 \Omega^4 + \Lambda_2 \Omega^2 + \Lambda_0}{\Delta_4 \Omega^4 + \Delta_2 \Omega^2 + \Delta_0}. \tag{A45}$$

**Appendix F. Expression for the Intrinsic Linewidth**

The laser intrinsic linewidth can be found from Equation (25). Defining:

$$\beta_{Ag} \equiv \frac{\tau_e^{-1} \zeta_S}{a_g G_0 S_0} \quad ; \quad \delta_{Ag} \equiv \frac{\zeta_S}{G_0} \quad ; \quad \Delta f_0 = \frac{R_{sp}}{4\pi S_0}, \tag{A46}$$

where, following [3]:

$$\delta_{Ag} \simeq 0 \simeq \beta_{Ag}, \tag{A47}$$

as $\delta_{Ag} < 10^{-2}$, which accounts for shot noise in the generation and recombination of minority carriers, and $\beta_{Ag}$ is inversely proportional to the laser power which above threshold becomes negligible. Starting from Equation (A45) and setting $\Omega = 0$ as required by Equation (25):

$$4\pi \Delta f = \Lambda_0 / \Delta_0$$

$$\overset{(A43c)(23)}{=} \frac{\left( \zeta_S \alpha_H a_g \right)^2 \left[ R_{sp} S_0 \left( 1 + \frac{(G_0)^2}{\zeta_S^2} + \frac{2G_0}{\zeta_S} \right) + N\tau_{sp}^{-1} \right] + \left( \tau_e^{-1} \zeta_S + a_g G_0 S_0 \right)^2 \frac{R_{sp}}{S_0}}{\left( a_g G_0 S_0 \mathsf{F}_3 + \mathsf{K}_c \zeta_S \tau_e^{-1} \right)^2}$$

$$\Leftrightarrow \frac{\Delta f}{\Delta f_0} = \frac{\left( \frac{\alpha_H}{G_0} \right)^2 \left( \zeta_S^2 + G_0^2 + 2G_0 \zeta_S + \frac{\zeta_S^2 N\tau_{sp}^{-1}}{R_{sp} S_0} \right) + \left( \beta_{Ag} + 1 \right)^2}{\left( \mathsf{K}_c \beta_{Ag} + \mathsf{F}_3 \right)^2}$$

$$= \frac{\left( \beta_{Ag} + 1 \right)^2 + \alpha_H^2 \left[ 1 + 2\delta_{Ag} + \delta_{Ag}^2 \left( 1 + \frac{N\tau_{sp}^{-1}}{R_{sp} S_0} \right) \right]}{\left( \mathsf{K}_c \beta_{Ag} + \mathsf{F}_3 \right)^2}$$

$$\overset{(A47)}{\Leftrightarrow} \frac{\Delta f}{\Delta f_0 \left(1 + \alpha_H^2\right)} \simeq F_3^{-2}$$

$$\overset{(23)(20)}{=} \{1 + \kappa_2 t_2 [\cos(\phi_2) - \alpha_H \sin(\phi_2)] - \kappa_1 t_1 [\cos(\phi_1) + \alpha_H \sin(\phi_1)]\}^{-2}$$

$$\overset{(A20a)(A20b)}{=} [1 + \gamma_H \kappa_2 t_2 \cos(\phi_2 + \theta_H) - \gamma_H \kappa_1 t_1 \cos(\phi_1 - \theta_H)]^{-2}. \tag{A48}$$

The presence of EOF from both sides of the laser cavity results in two terms in the linewidth expression, one for each side, as was seen in the threshold gain reduction and lasing frequency shift due to feedback. This result is discussed in Section 2.3.

## Appendix G. Suplementary Images: Tolerances

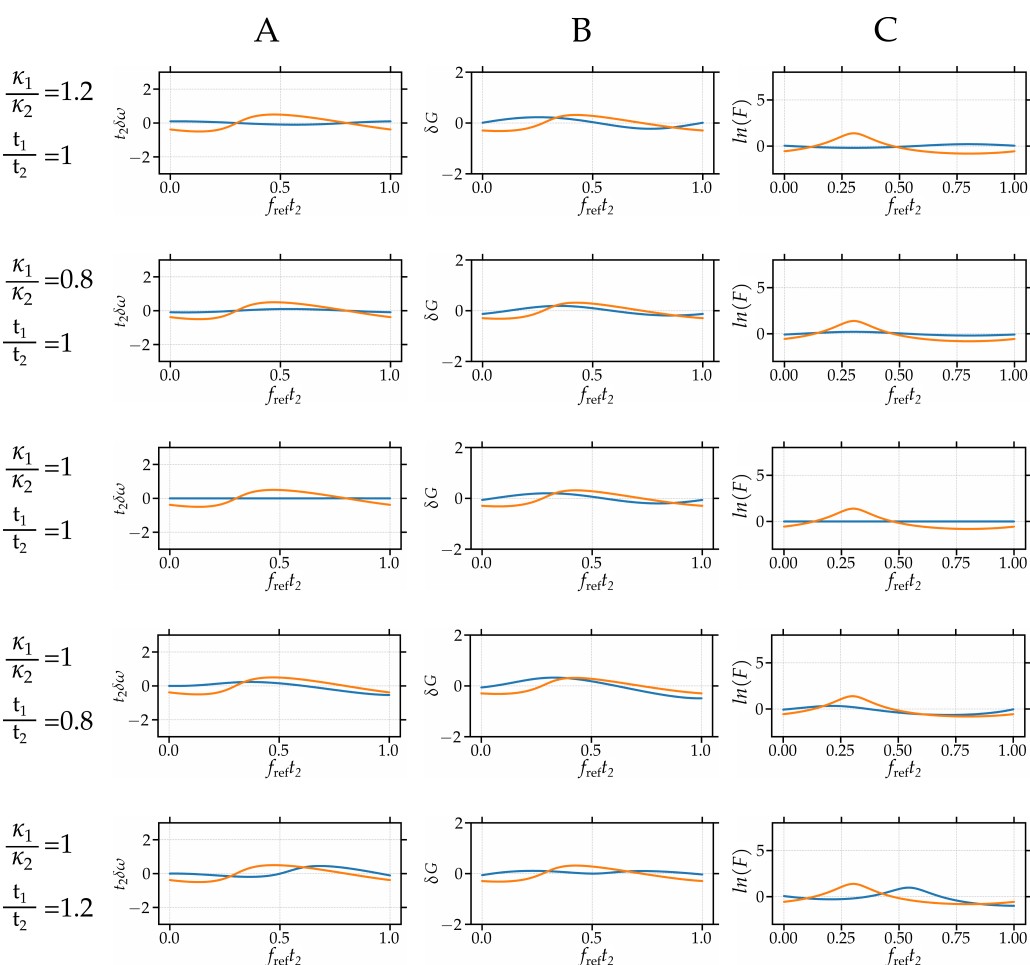

**Figure A2.** Numerical solutions for $C = 0.5$ under condition (12) for a $\pm 20\%$ variation of conditions (8) and (11). Double feedback case shown in blue, single feedback case shown in orange. Column (**A**) shows the lasing frequency shift results. Column (**B**) shows the threshold gain shift. Column (**C**) shows the intrinsic linewidth variations.

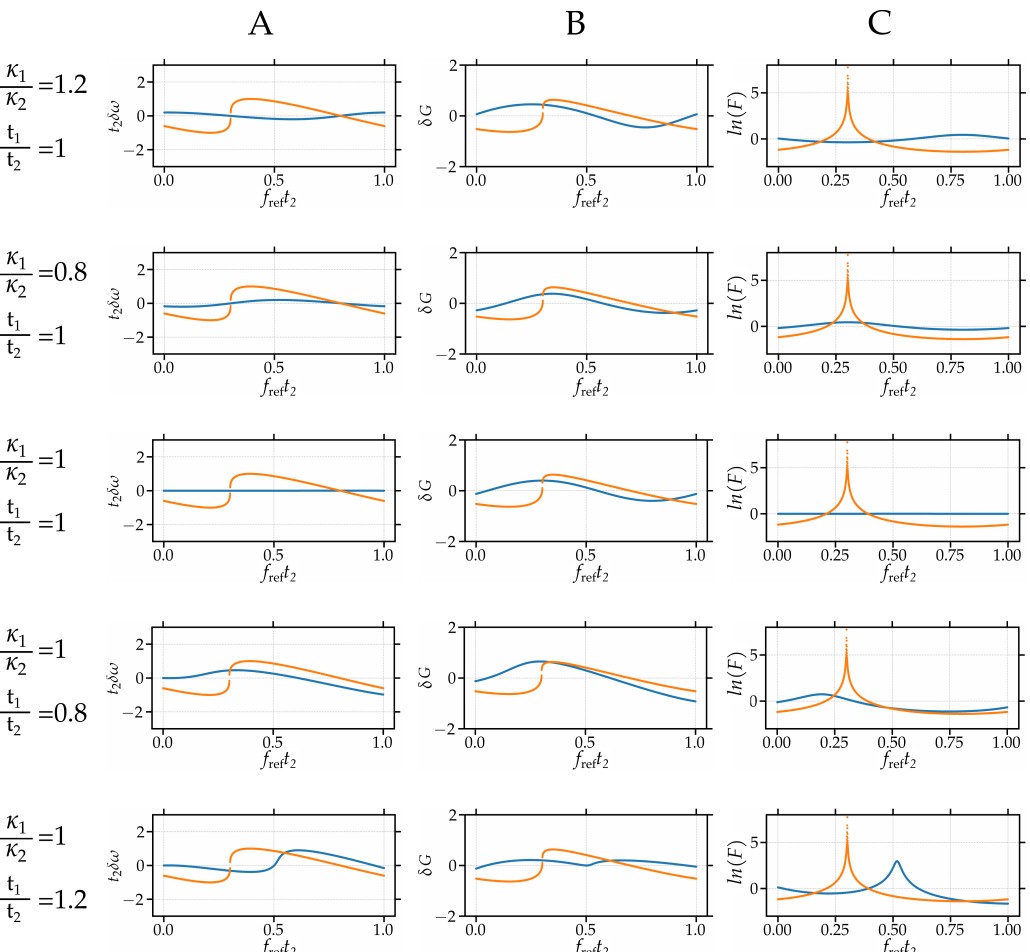

**Figure A3.** Numerical solutions for $C = 1$ under condition (12) for a $\pm 20\%$ variation of conditions (8) and (11). Double feedback case shown in blue, single feedback case shown in orange. Column (**A**) shows the lasing frequency shift results. Column (**B**) shows the threshold gain shift. Column (**C**) shows the intrinsic linewidth variations.

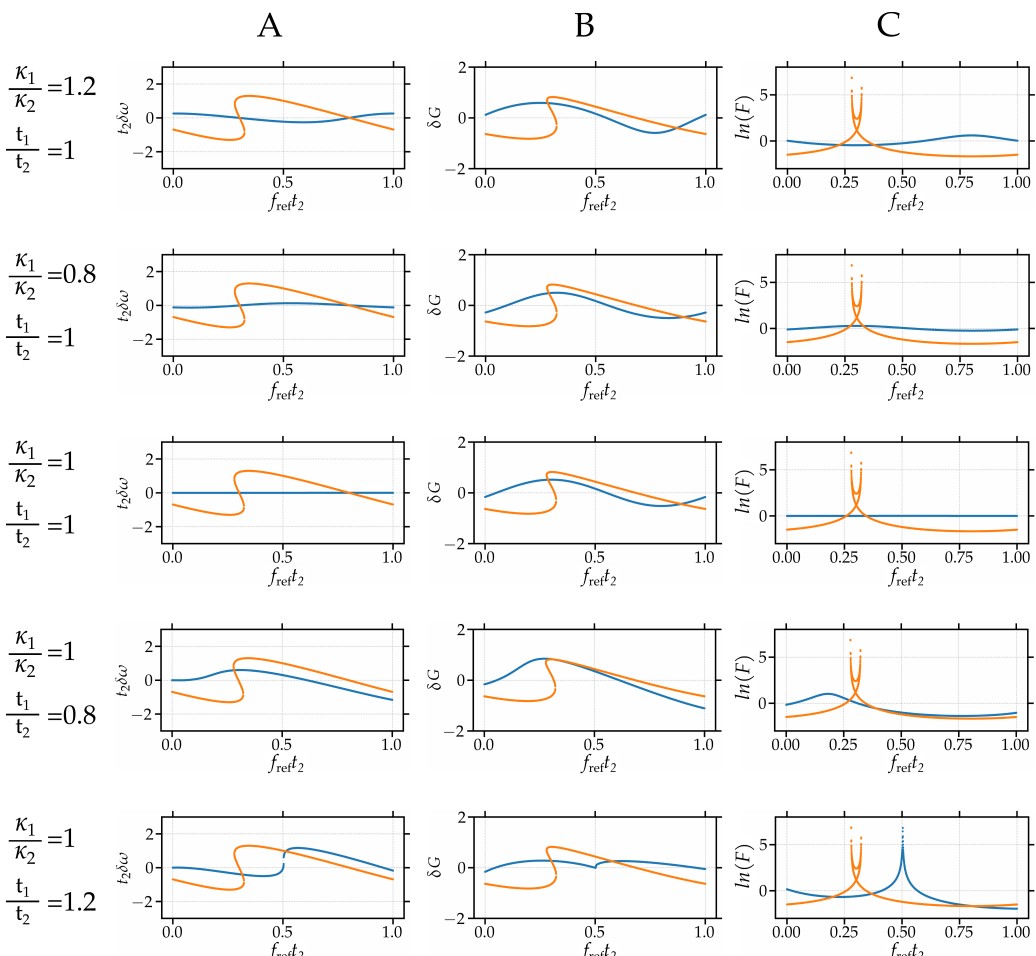

**Figure A4.** Numerical solutions for $C = 1.3$ under condition (12) for a $\pm 20\%$ variation of conditions (8) and (11). Double feedback case shown in blue, single feedback case shown in orange. Column (**A**) shows the lasing frequency shift results. Column (**B**) shows the threshold gain shift. Column (**C**) shows the intrinsic linewidth variations.

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
