# Peer review of "Dynamics of Semiconductor Lasers under External Optical Feedback from Both Sides of the Laser Cavity"

_photonics, doi:10.3390/photonics9010043_

Round 1

Reviewer 1 Report

The authors adopt EOF from both sides of the cavity to reduce and stabilize the laser linewidth and numerically simulate the lasing frequency shift, threshold gain shift and linewidth variations. The overall idea is interest to the research community and the corresponding derivation process as well as the simulation results are believable, but the following comments need to be addressed before acceptance.

  • The main idea of this paper is introducing EOF from both sides of the laser cavity, however, readers cannot catch the subject clearly from the title. Thus, it is advised to make some modifications on the title.
  • In Introduction, the authors should further clarify the novelty and advantages of their purposed scheme from the previous reported linewidth reduction work.
  • Too many equations are in Section 2, the authors are advised to reduce some unnecessary equations, such as the Eq. 2.22 to increase the readability.
  • The parameters for the given simulation results could be listed as a table.

Author Response

Thank you for your thoughtful comments and suggestions.Below we address (in black) your comments (in blue)

The main idea of this paper is introducing EOF from both sides of the laser cavity, however, readers cannot catch the subject clearly from the title. Thus, it is advised to make some modifications on the title.
Title has been modified following this suggestion.

In Introduction, the authors should further clarify the novelty and advantages of their purposed scheme from the previous reported linewidth reduction work.
This has been addressed by revising the introduction to emphasize this point (lines 60-66)

Too many equations are in Section 2, the authors are advised to reduce some unnecessary equations, such as the Eq. 2.22 to increase the readability.
We agree with this suggestion, thus several equations of minor importance have been removed from the text.

The parameters for the given simulation results could be listed as a table.
A table has been added to summarize the simulation parameter values.

Reviewer 2 Report

The authors study the dynamics of a semiconductor laser subject to double optical feedback from both sides of the laser.  Dynamics of semiconductor lasers with double external cavities has been reported in “Yun Liu and Junji Ohtsubo, Dynamics and chaos stabilization of semiconductor lasers with optical feedback, JQE, vol. 33. No. 7, 1163-1169, 1997”, which shows that there exist robust parameter regions for stable behaviors like fixed states or periodic oscillations. The linewidth is also greatly narrowed when fixed or periodic states are realized in the laser output with the introduction of the second feedback. Although both feedbacks are coupled into the same side of the laser, the results are similar to those reported in this paper. Authors should compare their results with this JQE paper in detail to improve the manuscript.

Author Response

Thank you for your highly relevant comment. The mentioned reference has been now discussed in the introduction (lines 33-40+44-48) and its equations compared to the ones in our manuscript. Results are difficult to compare, however, as a chaotic system is explored, and its response to the additional feedback term analyzed. This is different to our approach, where various values of feedback strength are studied, not limited to those that generate chaos. This has been commented on in the discussion section (lines 277-282)

Reviewer 3 Report

In the manuscript, the authors investigate the effect of the external optical feedback on the laser stability by considering the case of feedback on both sides of a laser. The detailed derivations of the formulas and equations are clearly presented in the manuscript. The physical quantities, i.e., lasing frequency, threshold gain and intrinsic linewidth, are numerically calculated to evaluate the laser stability. Different conditions are systematically analyzed. Overall, the manuscript is organized in a good manner and the main results are useful for the researchers working in the field. I recommend its publication in Photonics.

Minor suggestions:

--As the authors stated in the manuscript, an experimental study of this laser system is essential to validate the obtained results. I would suggest the authors to add a few sentences to comment on the previously published experimental results of a laser with single side feedback.

--I suggest the authors to carefully read the manuscript to correct some typos.

Author Response

Thank you for your thoughtful comments. Regarding the minor suggestions:

I would suggest the authors to add a few sentences to comment on the previously published experimental results of a laser with single side feedback.

A brief comment on this has been added at the end of the discussion section (lines 316-319)

I suggest the authors to carefully read the manuscript to correct some typos.
Thank you for pointing this out. We have revised the manuscript and corrected errors in the text. 

Reviewer 4 Report

The author proposed adding another cavity to the other side of the laser cavity. However, there are many issues need to be address before further consideration.

In Line 20, a bulk adding 13 refs in one line, this is not acceptable. 1-2 review works here is enough.

The idea of dual-cavity EOF system is not new, author should have reviews on some of the existing works such as the following,

  1. Zhu, Q. Chen, Y. Wang, H. Luo, H. Wu, and B. Ma, "Improvement on vibration measurement performance of laser self-mixing interference by using a pre-feedback mirror," Optics and Lasers in Engineering, vol. 105, pp. 150-158, 2018/06/01/ 2018.
  2. Ruan, B. Liu, Y. Yu, J. Xi, Q. Guo, and J. Tong, "High sensitive sensing by a laser diode with dual optical feedback operating at period-one oscillation," Applied Physics Letters, vol. 115, no.1, p. 011102, 2019.
  3. P. Mezzapesa, L. L. Columbo, M. Dabbicco, M. Brambilla, and G. Scamarcio, "QCL-based nonlinear sensing of independent targets dynamics," Optics Express, vol. 22, no. 5, pp. 5867-5874,2014/03/10 2014.

The rate equations and stable solutions are also being reported in these works, how does theirs different from the authors’?

There are too many equations in this paper, are these 2.19a,b,c the rate equations? What are the stable solutions? Authors should consider to list all used symbols into a table, there are many equations and symbols in this work, very hard to follow.

In figure2,3,4, what is the physical meaning for x axis (f0t2) in the simulation results?

In the section 3 numerical study, how do you increase the phase “deltaphi(m)” in practical?

In line 126 and Figure2,3,4 capture, what does “full system” means? Please give a definition before using it.

In line 128, please explain how did you observe there is no phase difference?

Line 129-130, please justify “narrowing for a wide range frequency”, how did you observed ?

I noted author said in line 214“higher levels of feedback strength are allowed without seeing multi valued solutions”, this is not really true when we looking at fig2,3. Clearly, under the full system, multi values still exist.

Line 243, the author claim, there is linewidth reduction are observed. This is not clearly demonstrated in the result, a numerical value should be presented when discussion the results.

The length of this paper is a problem, total 29 pages. It’s more like a book chapter rather than a journal paper. I don’t see the appendix has any real values in the paper. Also there are too many equations in sections 2. Authors should think about how to present the core value of the work, and make the paper more readable, concise, easy to follow the logic.

Author Response

Thank you for the useful comments to improve the quality of out manuscript. Below we answer (in black) your suggestions (in blue).

In Line 20, a bulk adding 13 refs in one line, this is not acceptable. 1-2 review works here is enough.
This has been addressed as suggested (line 20).

The idea of dual-cavity EOF system is not new, author should have reviews on some of the existing works.
Thank you for the additional reference suggestions. They have been cited and commented in the introduction (lines 40-48).

The rate equations and stable solutions are also being reported in these works, how does theirs different from the authors?
Rate equations are given in ref [Ruan 2019], an equation for the frequency shift is given in [Mezzapesa 2014] and [Zhu 2018], and a equation for the gain shift is given in [Mezapezza 2014]. These have now been compared with the ones in our work. (lines 89- 96 ;125-128); Having said that, none of the references include an analytical solution or analysis for laser linewidth, which is a main focus of our manuscript.

There are too many equations in this paper, are these 2.19a,b,c the rate equations? What are the stable solutions? Authors should consider to list all used symbols into a table, there are many equations and symbols in this work, very hard to follow.
Eq. 2.19 (now Eq. 2.16) are indeed the rate equations. This has been further clarified in the text. (line 122-123). The steady state solutions are mentioned in Appendix D, as only the main equations are kept in the main text, which is mentioned in line 128. The table was considered, but the introduction of new parameters is mostly condensed in a single page, thus the information of the table became too redundant, so we decided against including it.

In figure2,3,4, what is the physical meaning for x axis (f0t2) in the simulation results?
This has been revised and clarified in the text (Line 179-183).

In the section 3 numerical study, how do you increase the phase “deltaphi(m)” in practical?
Further clarification has been included in the text (Line 183-186)

In line 126 and Figure2,3,4 capture, what does “full system” means? Please give a definition before using it.
Thank you for pointing this out. ‘Full system’ refers to the double-feedback case proposed, and has accordingly been modified in the text.

In line 128, please explain how did you observe there is no phase difference?
That sentence refers to the case with  Δφ=0, which is one of the parameter dependences explored . It has been rewritten to avoid confusion. (Line 207-208)

Line 129-130, please justify “narrowing for a wide range frequency”, how did you observed ?
This has been clarified in Lines 203-204 ; 208-209

I noted author said in line 214“higher levels of feedback strength are allowed without seeing multi valued solutions”, this is not really true when we looking at fig2,3. Clearly, under the full system, multi values still exist.
Thanks for pointing this out. The sentence was meant to allude the cases near the stability conditions found, and has been revised accordingly (line 299-300).

Line 243, the author claim, there is linewidth reduction are observed. This is not clearly demonstrated in the result, a numerical value should be presented when discussion the results.
A numerical value is now included to support the claim (line 209-211 ; 324-326) 

The length of this paper is a problem, total 29 pages. It’s more like a book chapter rather than a journal paper. I don’t see the appendix has any real values in the paper. Also there are too many equations in sections 2. Authors should think about how to present the core value of the work, and make the paper more readable, concise, easy to follow the logic.
Several equations in the appendix have been removed, as well as equations in Sec. 2, reducing the total length of the paper.We do believe, however, that the appendix has value as it shows the detailed derivations of the formulas and equations that are analyzed in depth in the paper. Such derivations are expected to be useful and relevant for researchers that might want to explore optical feedback from both sides of the laser cavity, and as such we believe it should be kept.

Round 2

Reviewer 2 Report

The authors have addressed my concerns, the paper can be accepted for publication in Photonics.

Reviewer 4 Report

I'm satisfied with author's responses